# Paranoia as a deficit in non-social belief updating

Erin J Reed[1,2], Stefan Uddenberg[3], Praveen Suthaharan[4], Christoph D Mathys[5,6], Jane R Taylor[4], Stephanie Mary Groman[4], Philip R Corlett[4]*

[1]Interdepartmental Neuroscience Program, Yale School of Medicine, New Haven, United States; [2]Yale MD-PhD Program, Yale School of Medicine, New Haven, United States; [3]Princeton Neuroscience Institute, Princeton University, Princeton, United States; [4]Department of Psychiatry, Connecticut Mental Health Center, Yale University, New Have, United States; [5]Scuola Internazionale Superiore di Studi Avanzati (SISSA), Trieste, Italy; [6]Translational Neuromodeling Unit (TNU), Institute for Biomedical Engineering, University of Zurich and ETH Zurich, Zurich, Switzerland

*For correspondence:
philip.corlett@yale.edu

Competing interests: The authors declare that no competing interests exist.

**Abstract** Paranoia is the belief that harm is intended by others. It may arise from selective pressures to infer and avoid social threats, particularly in ambiguous or changing circumstances. We propose that uncertainty may be sufficient to elicit learning differences in paranoid individuals, without social threat. We used reversal learning behavior and computational modeling to estimate belief updating across individuals with and without mental illness, online participants, and rats chronically exposed to methamphetamine, an elicitor of paranoia in humans. Paranoia is associated with a stronger prior on volatility, accompanied by elevated sensitivity to perceived changes in the task environment. Methamphetamine exposure in rats recapitulates this impaired uncertainty-driven belief updating and rigid anticipation of a volatile environment. Our work provides evidence of fundamental, domain-general learning differences in paranoid individuals. This paradigm enables further assessment of the interplay between uncertainty and belief-updating across individuals and species.

## Introduction

Paranoia is excessive concern that harm will occur due to deliberate actions of others (*Freeman and Garety, 2000*). It manifests along a continuum of increasing severity (*Freeman et al., 2005*; *Freeman et al., 2010*; *Freeman et al., 2011*; *Bebbington et al., 2013*). Fleeting paranoid thoughts prevail in the general population (*Freeman, 2006*). A survey of over 7000 individuals found that nearly 20% believed people were against them at times in the past year; approximately 8% felt people had intentionally acted to harm them (*Freeman et al., 2011*). At a national level, paranoia may fuel divisive ideological intolerance. Historian Richard Hofstadter famously described catastrophizing, context insensitive political discourse as the 'paranoid style':

"The paranoid spokesman sees the fate of conspiracy in apocalyptic terms—he traffics in the birth and death of whole worlds, whole political orders, whole systems of human values. He is always manning the barricades of civilization. *He constantly lives at a turning point* [emphasis added]." (*Hofstadter, 1964*).

At its most severe, paranoia manifests as rigid beliefs known as delusions of persecution. These delusions occur in nearly 90% of first episode psychosis patients (*Freeman, 2007*). Psychostimulants also elicit severe paranoid states. Methamphetamine evokes new paranoid ideation particularly after repeated exposure or escalating doses (86% and 68%, respectively, in a survey of methamphetamine users) (*Leamon et al., 2010*).

**eLife digest** Everyone has had fleeting concerns that others might be against them at some point in their lives. Sometimes these concerns can escalate into paranoia and become debilitating. Paranoia is a common symptom in serious mental illnesses like schizophrenia. It can cause extreme distress and is linked with an increased risk of violence towards oneself or others. Understanding what happens in the brains of people experiencing paranoia might lead to better ways to treat or manage it.

Some experts argue that paranoia is caused by errors in the way people assess social situations. An alternative idea is that paranoia stems from the way the brain forms and updates beliefs about the world. Now, Reed et al. show that both people with paranoia and rats exposed to a paranoia-inducing substance expect the world will change frequently, change their minds often, and have a harder time learning in response to changing circumstances.

In the experiments, human volunteers with and without psychiatric disorders played a game where the best choices change. Then, the participants completed a survey to assess their level of paranoia. People with higher levels of paranoia predicted more changes would occur and made less predictable choices. In a second set of experiments, rats were put in a cage with three holes where they sometimes received sugar rewards. Some of the rats received methamphetamine, a drug that causes paranoia in humans. Rats given the drug also expected the location of the sugar reward would change often. The drugged animals had harder time learning and adapting to changing circumstances.

The experiments suggest that brain processes found in both rats, which are less social than humans, and humans contribute to paranoia. This suggests paranoia may make it harder to update beliefs. This may help scientists understand what causes paranoia and develop therapies or drugs that can reduce paranoia. This information may also help scientists understand why during societal crises like wars or natural disasters humans are prone to believing conspiracies. This is particularly important now as the world grapples with climate change and a global pandemic. Reed et al. note paranoia may impede the coordination of collaborative solutions to these challenging situations.

Paranoia has thus far defied explanation in mechanistic terms. Sophisticated Game Theory driven approaches (such as the Dictator Game [*Raihani and Bell, 2018*; *Raihani and Bell, 2017*]) have largely re-described the phenomenon — people who are paranoid have difficulties in laboratory tasks that require trust (*Raihani and Bell, 2019*). However, this is not driven by personal threat per se, but by negative representations of others (*Raihani and Bell, 2018*; *Raihani and Bell, 2017*). We posit that such representations are learned (*Fineberg et al., 2014*; *Behrens et al., 2008*), via the same fundamental learning mechanisms (*Cramer et al., 2002*) that underwrite non-social learning in non-human species (*Heyes and Pearce, 2015*). We hypothesize that aberrations to these domain-general learning mechanisms underlie paranoia. One such mechanism involves the judicious use of uncertainty to update beliefs: Expectations about the noisiness of the environment constrain whether we update beliefs or dismiss surprises as probabilistic anomalies. The higher the expected uncertainty (i.e., 'I expect variable outcomes'), the less surprising an atypical outcome may be, and the less it drives belief updates ('this variation is normal'). Unexpected uncertainty, in contrast, describes perceived change in the underlying statistics of the environment (*Yu and Dayan, 2005*; *Payzan-LeNestour and Bossaerts, 2011*; *Payzan-LeNestour et al., 2013*) (i.e. 'the world is changing'), which may call for belief revision.

Since excessive unexpected uncertainty is a signal of change, it might drive the recategorization of allies as enemies, which is a tenet of evolutionary theories of paranoia (*Raihani and Bell, 2019*). We tested the hypothesis that this drive to flexibly recategorize associations extends to non-social, domain-general inferences. We dissected learning mechanisms under expected and unexpected uncertainty – probabilistic variation and changes in underlying task structure (volatility). Here, volatility is a property of the task. Unexpected uncertainty is the perception of that volatility. Participants completed a non-social, three-option learning task which challenged them to form and revise associations between stimuli (colored card decks) and outcomes (points rewarded and lost), in addition to their beliefs about the volatility of the task environment. They encountered expected uncertainty as

probabilistic win or loss feedback ('each option yields positive and negative outcomes, but in different amounts'), and unexpected uncertainty as reassignment of reward probabilities between options ('sometimes the best option may change,' reversal events). Although reversal events elicit unexpected uncertainty by driving re-evaluation of the options, participants increasingly anticipate reversals and develop expectations about the stability of the task environment. We implemented an additional task manipulation: a shift in the underlying probabilities themselves (contingency transition, unsignaled to the participants), that effectively changes task volatility. Armed with the task structure and participants' choices, we applied a Hierarchical Gaussian Filter (HGF) model (*Mathys et al., 2011*; *Mathys et al., 2014*) which allowed us to infer participants' initial beliefs (i.e., priors) about task volatility, their readiness to learn about changes in the task volatility itself (meta-volatility learning rate) and learning rates that captured their expected and unexpected uncertainty regarding the task.

We examined the behavioral and computational correlates of paranoia both in-person and in a large online sample, spanning patients and healthy controls with varying degrees of paranoia. We also undertook a pre-clinical replication in rodents exposed chronically to saline or methamphetamine to determine whether a drug known to elicit paranoia in humans might induce similar perceptions of unexpected uncertainty, without contingency transition (*Groman et al., 2018*). We predicted that people with paranoia and rats administered methamphetamine would exhibit stronger priors on volatility, facilitating aberrant learning through unexpected uncertainty. We further hypothesized that this learning style would manifest as frequent and unnecessary choice switching (increased choice stochasticity and 'win-switch' behavior) rather than increased sensitivity to negative feedback (increased 'lose-switch' behavior/decreased 'lose-stay' behavior).

## Results

We analyzed belief updating across three reversal-learning experiments (*Figure 1*): an in laboratory pilot of patients and healthy controls, stratified by stable, paranoid personality trait (Experiment 1); four online task variants administered to participants via the Amazon Mechanical Turk (MTurk) marketplace (Experiment 2); and a re-analysis of data from rats on chronic, escalating doses of methamphetamine, a translational model of paranoia (Experiment 3) (*Groman et al., 2018*).

### Experiment 1

First, we explored trans-diagnostic associations between paranoia and reversal-learning in-person. Participants with and without psychiatric diagnoses (mood disorders: anxiety, depression, bipolar disorder, n = 8; schizophrenia spectrum: schizophrenia or schizoaffective disorder, n = 8; and healthy controls, n = 16), completed questionnaire versions of the *Structured Clinical Interview for DSM-IV Axis II Personality Disorders* (SCID-II) screening assessment (*Ryder et al., 2007*), Beck's Anxiety Inventory (BAI) (*Beck et al., 1988*), Beck's Depression Inventory (BDI) (*Beck et al., 1961*), and demographic assessments (*Table 1*). Approximately two-thirds of participants endorsed three or fewer items on the SCID-II paranoid personality subscale (median = 1 item). Participants who endorsed four or more items were classified as high paranoia (*n* = 11), consistent with the diagnostic threshold for paranoid personality disorder. Low paranoia (*n* = 21) and high paranoia groups did not differ significantly by age, nor were there significant group associations with gender, educational attainment, ethnicity, or race, although a larger percentage of paranoid participants identified as racial minorities or 'not specified' (*Table 1*). Diagnostic category (i.e., healthy control, mood disorder, or schizophrenia spectrum) was significantly associated with paranoia group membership, $\chi^2$ (2, *n* = 32)=12.329, p=0.002, Cramer's V = 0.621, as was psychiatric medication usage, $\chi^2$ (1, *n* = 32)=9.871, p=0.003, Cramer's V = 0.555. These differences were due to the higher proportion of healthy controls in the low paranoia group. As expected, paranoia, BAI, and BDI scores were significantly elevated in the high paranoia group relative to low paranoia controls (*Table 1*; paranoia: mean difference (MD) = 0.536, CI=[0.455,0.618], *t*(30)=13.476, p=2.92E-14, Hedges' *g* = 5.016; BAI: MD = 0.585, CI=[0.239, 0.931], *t*(30)=3.453, p=0.002, Hedges' *g* = 1.285, MD = −0.585; BDI: MD = 0.427, CI= [0.078, 0.775], *t*(11.854) = 2.67, p=0.021, Hedges' *g* = 1.255).

Participants completed a three-option reversal-learning task in which they chose between three decks of cards with hidden reward probabilities (*Figure 1a and b*). They selected a deck on each turn and received positive or negative feedback (+100 or −50 points, respectively). They were

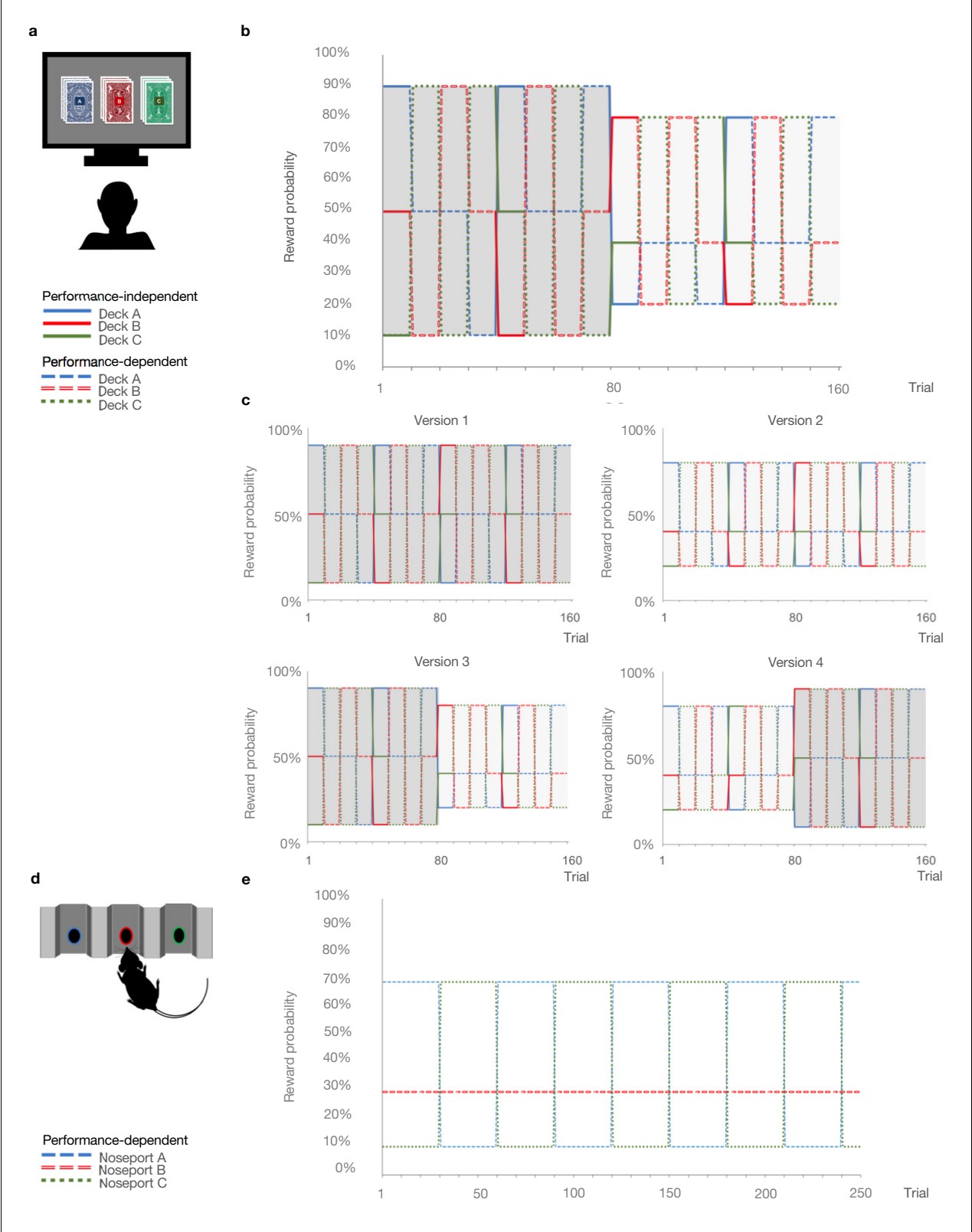

**Figure 1.** Probabilistic reversal learning task. (a) Human paradigm: participants choose between three decks of cards with different colored backs (Blue, Red, and Green) with different, unknown probabilities of reward and loss. (b) Reward contingency schedule for in laboratory experiment (Reward probabilities associated with the different colored decks, Blue, Red, Green, across trials and blocks). On trial 81, the probability context shifted from 90%, 50%, and 10% (dark grey) to 80%, 40%, and 20% without warning (light grey). (c), Reward contingency schedules for online experiment. (d) Rat

*Figure 1 continued on next page*

*Figure 1 continued*

paradigm: subjects choose between three noseports (Blue, Red, Green, for illustrative pupuses) with different probabilities of sucrose pellet reward. (e) Reward contingency schedule for rat experiment (*Groman et al., 2018*). Performance dependent reversals occur after a certain number of choices of the high reward deck. Performance independent reversals occur regardless of participant behavior.

instructed to find the best deck with the caveat that the best deck may change. Undisclosed to participants, reward probabilities switched among decks after selection of the highest probability option in nine out of ten consecutive trials ('reversal events'). Thus, the task was designed to elicit expected uncertainty (probabilistic reward associations) and unexpected uncertainty (reversal events), requiring participants to distinguish probabilistic losses from change in the underlying deck values. In addition, reward contingencies changed from 90%, 50%, and 10% chance of reward to 80%, 40%, and 20% between the first and second halves of the task ('contingency transition'; block 1 = 80 trials, 90-50–10%; block 2 = 80 trials, 80-40–20%, unsignaled to the participants). This transition altered the volatility of the task environment, thereby making it more difficult to achieve reversals and often delaying their occurrence. Successful achievement of reversals was contingent upon adapting stay-vs-switch strategies, thereby testing subjects' abilities to update beliefs about the overall task volatility ('metavolatility learning'). High paranoia subjects achieved fewer reversals (MD = $-2.31$, CI=$[-4.504, -0.111,]$, $t(30)$=-2.145, p=0.04, Hedges' $g$ = 0.798), but total points earned did not significantly differ, suggesting that there was no penalty for the different behaviors expressed by the more paranoid subjects (*Table 1*). We predicted that paranoia would be associated with unexpected uncertainty-driven belief updating.

## Experiment 2

We aimed to replicate and extend our investigation of paranoia and reversal-learning in a larger online sample. We administered three alternative task versions to control for the contingency transition (*Figure 1c*). Version 1 ($n$ = 45 low paranoia, 20 high paranoia) provided a constant contingency of 90-50–10% reward probabilities (Easy-Easy); version 2 ($n$ = 69 low paranoia, 18 high paranoia) provided a constant contingency of 80-40–20% (Hard-Hard); version 3 ($n$ = 56 low paranoia, 16 high paranoia) served to replicate Experiment 1 with a contingency transition from 90-50–10% to 80-40–20% (Easy-Hard); version 4 ($n$ = 64 low paranoia, 19 high paranoia) provided the reverse contingency transition, 80-40–20% to 90-50–10% (Hard-Easy). The stable contingencies (versions 1 and 2) lacked contingency transitions. Versions 3 and 4 manipulated task volatility mid-way, although the contingency transition was not signalled to participants. We predicted that high paranoia participants would find versions 3 and 4 particularly challenging. Given that version 3 is easier to learn initially, we expected participants to develop stronger priors and thus be more confounded by the contingency transition, compared to version four participants.

Participants' demographic and mental health questionnaire responses did not differ significantly across task version experiments (*Table 2*). Total points and reversals achieved suggest variations in task difficulty (*Table 2*, version effects: points earned, $F(3, 299)$=32.288, p=4.16E-18, $\eta_p^2$=0.245; reversals achieved, $F(3, 299)$=4.329, p=0.005, $\eta_p^2$=0.042), but there was no significant association between task version and attrition rate (52.7%, 52.9%, 54.6%, and 53.1% attrition, respectively; $\chi^2$ (3, n = 752)=0.167, p=0.983, Cramer's V = 0.015).

Across task versions, high paranoia participants endorsed higher BAI and BDI scores ($n$ = 73 high paranoia, 234 low paranoia; BAI: $F(1, 299)$=38.752, p=1.63E-09, $\eta_p^2$=0.115; BDI: $F(1, 299)$=74.528, p=3.62E-16, $\eta_p^2$=0.20; *Table 2*). Both correlated with paranoia (BAI: Pearson's $r$ = 0.450, p=1.09E-16, CI=[0.348, 0.55]; BDI: Pearson's $r$ = 0.543, p=6.26E-25, CI=[0.448, 0.638]). Trial-by-trial reaction time did not differ significantly between low and high paranoia (*Table 2*), but high paranoia participants earned fewer total points ($F(1, 299)$=6.175, p=0.014, $\eta_p^2$=0.020) and achieved fewer reversals ($F(1, 299)$=5.762, p=0.017, $\eta_p^2$=0.019; *Table 2*). Deck choice perseveration after negative feedback (lose-stay behavior) did not significantly differ by paranoia group, but choice switching after positive feedback (win-switch behavior) was elevated in high paranoia (block 1: $F(1, 299)$=7.117, p=0.008, $\eta_p^2$=0.023; block 2: $F(1, 299)$=9.918, p=0.002, $\eta_p^2$=0.032; *Table 2*).

**Table 1.** In Lab vs. Online Version 3.

| | In Lab | | | | Online Version 3 | | | |
|---|---|---|---|---|---|---|---|---|
| | Low Paranoia (n=21) | High Paranoia (n=11) | Statistic | p-value | Low Paranoia (n=56) | High Paranoia (n=16) | Statistic | p-value |
| **Demographics** | | | | | | | | |
| Age (years) | 36.0 [3.2] | 38.9 [3.9] | -0.531 (27)† | 0.6 | 38.6 [1.6] | 32.9 [1.7] | 2.441 (41.8)† | **0.019**[¶] |
| Gender | | | 0.006 (1)‡ | 1[§] | | | .780 (1)‡ | 0.410 |
| % Female | 71.4% | 72.7% | n/a | n/a | 50.0% | 62.5% | n/a | n/a |
| % Male | 28.6% | 27.3% | n/a | n/a | 50.0% | 37.5% | n/a | n/a |
| % Other or not specified | 0% | 0% | n/a | n/a | 0% | 0% | n/a | n/a |
| Education | | | 4.972 (6)‡ | 0.638[§] | | | 5.351 (6)‡ | 0.549[§] |
| % High school degree or equivalent | 19.0% | 45.5% | n/a | n/a | 16.1% | 6.3% | n/a | n/a |
| % Some college or university, no degree | 14.3% | 0% | n/a | n/a | 17.9% | 25.0% | n/a | n/a |
| % Associate degree | 9.5% | 9.1% | n/a | n/a | 12.5% | 12.5% | n/a | n/a |
| % Bachelor's degree | 23.8% | 27.3% | n/a | n/a | 35.7% | 56.3% | n/a | n/a |
| % Master's degree | 9.5% | 0% | n/a | n/a | 14.3% | 0% | n/a | n/a |
| % Doctorate or professional degree | 4.8% | 0% | n/a | n/a | 1.8% | 0% | n/a | n/a |
| % Completed some postgraduate | 0% | 0% | n/a | n/a | 1.8% | 0% | n/a | n/a |
| % Other / not specified | 19.0% | 18.2% | n/a | n/a | 0% | 0% | n/a | n/a |
| Ethnicity | | | .134 (1)‡ | 1[§] | | | .117 (1)‡ | 1[§] |
| % Hispanic, Latino, or Spanish origin | 23.8% | 18.2% | n/a | n/a | 8.9% | 6.3% | n/a | n/a |
| % Not of Hispanic, Latino, or Spanish origin | 76.2% | 81.8% | n/a | n/a | 91.1% | 93.8% | n/a | n/a |
| Race | | | 6.250 (4)‡ | 0.186[§] | | | 5.368 (4)‡ | 0.229[§] |
| % White | 61.9% | 36.4% | n/a | n/a | 85.7% | 75.0% | n/a | n/a |
| % Black or African American | 19.0% | 36.4% | n/a | n/a | 0% | 12.5% | n/a | n/a |
| % Asian | 14.3% | 9.1% | n/a | n/a | 3.6% | 6.3% | n/a | n/a |
| % American Indian or Alaska Native | 4.8% | 0% | n/a | n/a | 1.8% | 6.3% | n/a | n/a |
| % Multiracial | 0% | 0% | n/a | n/a | 3.6% | 0% | n/a | n/a |
| % Other / not specified | 0% | 18.2% | n/a | n/a | 5.4% | 0% | n/a | n/a |
| **Mental Health** | | | | | | | | |
| Psychiatric diagnosis | | | 12.329 (2)‡ | **0.002**[§] | | | 7.850 (3)‡ | **0.039**[§] |
| % No psychiatric diagnosis | 71.4% | 9.1% | adj. residuals | **0.004** | 71.4% | 50.0% | adj. residuals | 0.465 |
| % Schizophrenia spectrum | 19.0% | 36.4% | adj. residuals | 0.546 | 0% | 6.3% | adj. residuals | 0.307 |
| % Mood disorder | 9.5% | 54.5% | adj. residuals | 0.020[#] | 21.4% | 43.8% | adj. residuals | 0.356 |
| % Not specified | 0% | 0% | adj. residuals | n/a | 7.1% | 0% | adj. residuals | 0.751 |
| % Medicated | 23.8% | 81.8% | 9.871 (1)‡ | **0.003**[§] | 7.1% | 31.3% | 8.730 (2)‡ | **0.023**[§] |
| Beck's Anxiety Inventory | 0.27 [0.08] | 0.85 [0.17] | -3.453 (30)† | **0.002** | 0.24 [0.04] | 0.90 [0.20] | -3.303 (16.179)† | **0.004**[¶] |
| Beck's Depression Inventory | 0.23 [0.05] | 0.66 [0.15] | -2.67 (11.854)† | **0.021**[¶] | 0.25 [0.04] | 1.03 [0.19] | -3.951 (16.659)† | **0.001**[¶] |
| SCID Paranoia Personality Score | 0.09 [0.02] | 0.63 [0.04] | -13.476 (30)† | **2.92E-14** | 0.1 [0.02] | 0.72 [0.04] | -16.551 (70)† | **6.712E-26** |
| **Reversal Learning Performance** | | | | | | | | |
| Total points earned | 7061.9 [286.9] | 6290.9 [372.2] | 1.608 (30)† | 0.118 | 7533.0 [143.8] | 6503.1 [340.6] | 3.177 (70)† | **0.002** |
| Total reversals achieved | 4.8 [0.7] | 2.5 [0.8] | 2.145 (30)† | **0.04** | 6.3 [0.3] | 4.9 [0.8] | 1.758 (20.14)† | 0.094[¶] |
| % Achieving reversals | 90.5% | 72.7% | 1.407 (1)‡ | 0.327[§] | 100% | 87.5% | 7.200 (1)‡ | **0.047**[§] |
| Trials to first reversal | 29.2 [4.5] | 27.9 [11] | 0.136 (25)† | 0.893 | 20.0 [1.7] | 13.7 [1.8] | 1.774 (68)† | 0.081 |
| % Recovering post-reversal | 81.0% | 54.5% | 2.490 (1)‡ | 0.213[§] | 91.1% | 69.0% | 3.482 (1)‡ | 0.097[§] |
| Trials to switch | 1.68 [0.22] | 1.43 [0.20] | 0.671 (24)† | 0.509 | 2.1 [0.2] | 2.6 [0.6] | -1.088 (64)† | 0.280 |

*Table 1 continued on next page*

Table 1 continued

| | In Lab | | | | Online Version 3 | | | |
| --- | --- | --- | --- | --- | --- | --- | --- | --- |
| | Low Paranoia (n=21) | High Paranoia (n=11) | Statistic | p-value | Low Paranoia (n=56) | High Paranoia (n=16) | Statistic | p-value |
| *Trials to recovery* | 3.75 [0.51] | 4 [0.93] | -0.285 (21)† | 0.779 | 2.9 [0.3] | 4.9 [0.8] | -2.694 (60)† | **0.009** |
| Win-switch rate, block 1 (90-50-10) | 0.08 [0.03] | 0.24 [0.09] | -1.742 (12.379)† | 0.106¶ | 0.04 [0.01] | 0.13 [0.05] | -1.906 (15.762)† | 0.075¶ |
| Win-switch rate, block 2 (80-40-20) | 0.07 [0.04] | 0.21 [0.1] | -1.601 (30)† | 0.12 | 0.02 [0.01] | 0.12 [0.05] | -2.02 (15.915)† | 0.061¶ |
| Lose-stay rate, block 1 (90-50-10) | 0.19 [0.03] | 0.13 [0.06] | 0.919 (30)† | 0.365 | 0.30 [0.03] | 0.39 [0.06] | -1.425 (70)† | 0.158 |
| Lose-stay rate, block 2 (80-40-20) | 0.26 [0.05] | 0.12 [0.05] | 1.817 (30)† | 0.079 | 0.33 [0.03] | 0.37 [0.06] | -0.554 (70)† | 0.581 |
| Null trials | 8.5 [2.8] | 10.4 [3.7] | -0.391 (30)† | 0.699 | n/a | n/a | n/a | n/a |

† Independent samples t-test: t-value (df). Two-tailed p-values reported ‡ Exact test, chi-square coefficient (df)§ Exact significance (2-sided)¶ Equal variances not assumed # Not significant (bonferonni correction).

## Experiment 3

To translate across species, we performed a new analysis of published data from rats exposed to chronic methamphetamine (*Groman et al., 2018*). Rats chose between three operant chamber noseports with differing probabilities of sucrose reward (70%, 30%, and 10%; *Figure 1d and e*). Contingencies switched between the 70% and 10% noseports after selection of the highest reinforced option in 21 out of 30 consecutive trials (*Figure 1e*). This task was most similar in structure to the first blocks of online versions 2 and 4. There was no increase in unexpected volatility mid-way through the task. Rats were tested for 26 within-session reversal blocks (Pre-Rx, *n* = 10 per group), administered saline or methamphetamine according to a 23 day schedule mimicking the escalating doses and frequencies of chronic human methamphetamine users (*Groman et al., 2018*), and tested once per week for four weeks following completion of the drug regimen (Post-Rx; *n* = 10 saline, seven methamphetamine) (*Groman et al., 2018*). Relative to rats exposed to saline, those rats exposed to methamphetamine exhibited increased win-switch behavior, similar to what we has observed in the high paranoia human participants, and additionally, unlike humans, they perseverated after negative feedback (*Groman et al., 2018*).

## Computational modeling

We employed hierarchical Gaussian filter (HGF) modeling to compare belief updating across individuals with low and high paranoia, as well as across human participants and rats exposed to methamphetamine (*Table 3*). We paired a three-level perceptual model with a softmax decision model dependent upon third level volatility (*Figure 2a*). We inverted the model from subject data (trial-by-trial choices and feedback) to estimate parameters for each individual (*Figure 2b*). Level 1 ($x_1$) characterizes trial-by-trial perception of task feedback (win or loss in humans, reward or no reward in rats), Level 2 ($x_2$) distinguishes stimulus-outcome associations (deck or noseport values), and Level 3 ($x_3$) renders perception of the overall task volatility (i.e., frequency of reversal events, changes in the stimulus-outcome associations).

Belief trajectories were unique to each subject due to the probabilistic, performance-dependent nature of the task, so we estimated initial beliefs (priors) for $x_2$ and $x_3$ ($\mu_2^0$ and $\mu_3^0$, respectively). We also estimated $\omega_2$, the tonic volatility of stimulus-outcome associations. Lower $\omega_2$ indicates that subjects are slower to adjust beliefs about the value of each option; they maintain rigid beliefs about the underlying probabilities. The $\kappa$ parameter captures the impact of phasic volatility on updating stimulus-outcome associations. In the setting of our experiments, $\kappa$ approximates the influence of unexpected uncertainty. Higher $\kappa$ implies faster updating of stimulus-outcome associations – that is, participants are more likely perceive volatility as reversal events. Our final parameter of interest, $\omega_3$, characterizes perception of 'meta-volatility,' such as changes in the frequency of reversal events (*Lawson et al., 2017*). The lower $\omega_3$, the slower a subject is to adjust their volatility belief; they adhere more rigidly to their volatility prior ($\mu_3^0$).

Priors did not differ between groups at $x_2$ (*Table 3*) but paranoid individuals and rats exposed to methamphetamine exhibited elevated $\mu_3^0$, they expected greater task volatility (*Figure 2b*, blue). In

**Table 2.** Online experiment.

| | Version 1 | | Version 2 | | Version 3 | | Version 4 | | Version Effect | | Paranoia Effect | | Interaction | |
|---|---|---|---|---|---|---|---|---|---|---|---|---|---|---|
| | Low Paranoia (n=45) | High Paranoia (n=20) | Low Paranoia (n=69) | High Paranoia (n=18) | Low Paranoia (n=56) | High Paranoia (n=16) | Low Paranoia (n=64) | High Paranoia (n=19) | Statistic | p-value | Statistic | p-value | Statistic | p-value |
| **Demographics** | | | | | | | | | | | | | | |
| Age (years) | 36.5 [1.5] | 35.4 [2.4] | 36.2 [1.4] | 39.5 [2.8] | 38.6 [1.6] | 32.9 [1.7] | 37.6 [1.3] | 30.7 [1.6] | 1.12 (3)[††] | 0.342 | 3.202 (1)[††] | 0.075 | 2.619 (3)[††] | 0.051 |
| Gender | | | | | | | | | 7.29 (6)[‡] | 0.238[§] | 1.373 (2)[‡] | 0.503[§] | n/a | n/a |
| % Female | 44.4% | 45.0% | 47.8% | 50.0% | 50.0% | 62.5% | 57.8% | 73.7% | n/a | n/a | n/a | n/a | n/a | n/a |
| % Male | 55.6% | 55.0% | 50.7% | 50.0% | 50.0% | 37.5% | 42.2% | 26.3% | n/a | n/a | n/a | n/a | n/a | n/a |
| % Other or not specified | 0% | 0% | 1.4% | 0% | 0% | 0% | 0% | 0% | n/a | n/a | n/a | n/a | n/a | n/a |
| Education | | | | | | | | | 15.9 (21)[‡] | 0.812[‖] | 7.326 (7)[‡] | 0.4[§] | n/a | n/a |
| % High school degree or equivalent | 17.8% | 20.0% | 13.0% | 16.7% | 16.1% | 6.3% | 25.0% | 10.5% | n/a | n/a | n/a | n/a | n/a | n/a |
| % Some college or university, no degree | 22.2% | 30.0% | 24.6% | 22.2% | 17.9% | 25.0% | 25.0% | 26.3% | n/a | n/a | n/a | n/a | n/a | n/a |
| % Associate degree | 13.3% | 15.0% | 17.4% | 22.2% | 12.5% | 12.5% | 9.4% | 21.1% | n/a | n/a | n/a | n/a | n/a | n/a |
| % Bachelor's degree | 33.3% | 35.0% | 40.6% | 22.2% | 35.7% | 56.3% | 28.1% | 31.6% | n/a | n/a | n/a | n/a | n/a | n/a |
| % Master's degree | 8.9% | 0% | 2.9% | 0% | 14.3% | 0% | 7.8% | 10.5% | n/a | n/a | n/a | n/a | n/a | n/a |
| % Doctorate or professional degree | 4.4% | 0% | 0% | 5.6% | 1.8% | 0% | 1.6% | 0% | n/a | n/a | n/a | n/a | n/a | n/a |
| % Completed some postgraduate | 0% | 0% | 1.4% | 5.6% | 1.8% | 0% | 3.1% | 0% | n/a | n/a | n/a | n/a | n/a | n/a |
| % Other / not specified | 0% | 0% | 0% | 5.6% | 0% | 0% | 0% | 0% | n/a | n/a | n/a | n/a | n/a | n/a |
| Income | | | | | | | | | 14.961 (18)[‡] | .671[‖] | 1.177 (6)[‡] | 0.981[§] | n/a | n/a |
| Less than $20,000 | 24.4% | 25.0% | 24.6% | 33.3% | 17.9% | 37.5% | 23.4% | 15.8% | n/a | n/a | n/a | n/a | n/a | n/a |
| $20,000 to $34,999 | 40.0% | 25.0% | 20.3% | 22.2% | 33.9% | 31.3% | 28.1% | 31.6% | n/a | n/a | n/a | n/a | n/a | n/a |
| $35,000 to $49,999 | 15.6% | 15.0% | 18.8% | 16.7% | 12.5% | 6.3% | 18.8% | 15.8% | n/a | n/a | n/a | n/a | n/a | n/a |
| $50,000 to $74,999 | 13.3% | 35.0% | 20.3% | 5.6% | 21.4% | 12.5% | 18.8% | 21.1% | n/a | n/a | n/a | n/a | n/a | n/a |
| $75,000 to $99,999 | 4.4% | 0% | 7.2% | 11.1% | 8.9% | 6.3% | 7.8% | 15.8% | n/a | n/a | n/a | n/a | n/a | n/a |
| Over $100,000 | 0% | 0% | 5.8% | 5.6% | 3.6% | 6.3% | 1.6% | 0% | n/a | n/a | n/a | n/a | n/a | n/a |
| Not specified | 2.2% | 0% | 2.9% | 5.6% | 1.8% | 0% | 1.6% | 0% | n/a | n/a | n/a | n/a | n/a | n/a |
| Cognitive Reflection | | | | | | | | | 11.922 (9)[‡] | 0.223[‖] | 7.002 (3)[‡] | 0.071[§] | n/a | n/a |
| % Answering 0/3 correctly | 11.1% | 25.0% | 10.1% | 11.1% | 17.9% | 25.0% | 15.6% | 26.3% | n/a | n/a | n/a | n/a | n/a | n/a |
| % Answering 1/3 correctly | 4.4% | 5.0% | 15.9% | 11.1% | 8.9% | 25.0% | 14.1% | 15.8% | n/a | n/a | n/a | n/a | n/a | n/a |
| % Answering 2/3 correctly | 13.3% | 25.0% | 15.9% | 16.7% | 19.6% | 25.0% | 21.9% | 31.6% | n/a | n/a | n/a | n/a | n/a | n/a |
| % Answering 3/3 correctly | 71.1% | 45.0% | 58.0% | 61.1% | 53.6% | 25.0% | 48.4% | 26.3% | n/a | n/a | n/a | n/a | n/a | n/a |
| Ethnicity | | | | | | | | | 5.162 (3)[‡] | 0.157[§] | 3.715 (1)[‡] | 0.069[§] | n/a | n/a |

*Table 2 continued on next page*

Table 2 continued

| | Version 1 | | Version 2 | | Version 3 | | Version 4 | | Version Effect | | Paranoia Effect | | Interaction | |
|---|---|---|---|---|---|---|---|---|---|---|---|---|---|---|
| | Low Paranoia (n=45) | High Paranoia (n=20) | Low Paranoia (n=69) | High Paranoia (n=18) | Low Paranoia (n=56) | High Paranoia (n=16) | Low Paranoia (n=64) | High Paranoia (n=19) | Statistic | p-value | Statistic | p-value | Statistic | p-value |
| % Hispanic, Latino, or Spanish origin | 4.4% | 15.0% | 1.4% | 0% | 8.9% | 6.3% | 1.6% | 15.8% | n/a | n/a | n/a | n/a | n/a | n/a |
| % Not of Hispanic, Latino, or Spanish origin | 95.6% | 85.0% | 98.6% | 100.0% | 91.1% | 93.8% | 98.4% | 84.2% | n/a | n/a | n/a | n/a | n/a | n/a |
| Race | | | | | | | | | 19.559 (15)‡ | .173|| | 9.626 (5)‡ | 0.084§ | n/a | n/a |
| % White | 82.2% | 75.0% | 84.1% | 88.9% | 85.7% | 75.0% | 85.9% | 73.7% | n/a | n/a | n/a | n/a | n/a | n/a |
| % Black or African American | 6.7% | 15.0% | 5.8% | 11.1% | 0% | 12.5% | 4.7% | 10.5% | n/a | n/a | n/a | n/a | n/a | n/a |
| % Asian | 8.9% | 10.0% | 7.2% | 0% | 3.6% | 6.3% | 7.8% | 0% | n/a | n/a | n/a | n/a | n/a | n/a |
| % American Indian or Alaska Native | 0% | 0% | 0% | 0% | 1.8% | 6.3% | 0% | 0% | n/a | n/a | n/a | n/a | n/a | n/a |
| % Multiracial | 2.2% | 0% | 1.4% | 0% | 3.6% | 0% | 1.6% | 15.8% | n/a | n/a | n/a | n/a | n/a | n/a |
| % Other / not specified | 0% | 0% | 1.4% | 0% | 5.4% | 0% | 0% | 0% | n/a | n/a | n/a | n/a | n/a | n/a |
| **Mental Health** | | | | | | | | | | | | | | |
| Psychiatric diagnosis | | | | | | | | | 10.783 (9)‡ | 0.292|| | 2.960 (3)‡ | 0.361§ | n/a | n/a |
| % No psychiatric diagnosis | 73.3% | 80.0% | 60.9% | 55.6% | 71.4% | 50.0% | 65.6% | 42.1% | n/a | n/a | n/a | n/a | n/a | n/a |
| % Schizophrenia spectrum | 2.2% | 0% | 0% | 0% | 0% | 6.3% | 0% | 0% | n/a | n/a | n/a | n/a | n/a | n/a |
| % Mood disorder | 13.3% | 15.0% | 27.5% | 22.2% | 21.4% | 43.8% | 26.6% | 31.6% | n/a | n/a | n/a | n/a | n/a | n/a |
| % Not specified | 11.1% | 5.0% | 11.6% | 22.2% | 7.1% | 0% | 7.8% | 26.3% | n/a | n/a | n/a | n/a | n/a | n/a |
| % Medicated | 8.9% | 10.0% | 13.0% | 22.2% | 7.1% | 31.3% | 14.1% | 10.5% | 3.575 (6)‡ | 0.744§ | 4.164 (2)‡ | 0.121§ | n/a | n/a |
| Beck's Anxiety Inventory | 0.34 [0.06] | 0.52 [0.14] | 0.31 [0.04] | 0.6 [0.13] | 0.24 [0.04] | 0.90 [0.20] | 0.33 [0.06] | 0.79 [0.18] | 1.244 (3)† | 0.2941 | 38.752 (1)†† | 1.63E-09 | 2.577 (3)†† | 0.0539 |
| Beck's Depression Inventory | 0.36 [0.07] | 0.86 [0.15] | 0.32 [0.05] | 0.79 [0.13] | 0.25 [0.04] | 1.03 [0.19] | 0.38 [0.07] | 1.06 [0.20] | 1.023 (3)† | 0.3827 | 74.528 (1)†† | 3.62E-16 | 1.089 (3)†† | 0.3542 |
| SCID Paranoia Personality Score | 0.11 [0.02] | 0.67 [0.04] | 0.11 [0.02] | 0.61 [0.03] | 0.1 [0.02] | 0.72 [0.04] | 0.11 [0.02] | 0.65 [0.03] | 1.297 (3)† | 0.2756 | 879.379 (1)†† | 4.81E-91 | 2.018 (3)†† | 0.1114 |
| **Reversal Learning Performance** | | | | | | | | | | | | | | |
| Total points earned | 8656.7 [182.9] | 8372.5 [405.2] | 6045.7 [135.7] | 6266.7 [288.0] | 7533.0 [143.8] | 6503.1 [340.6] | 7171.1 [1175.6] | 6510.5 [403.6] | 32.288 (3)† | 4.16E-18 | 6.175 (1)†† | 0.0135 | 2.258 (3)†† | 0.0818 |
| Total reversals achieved | 7.2 [0.3] | 6.5 [0.5] | 5.5 [0.3] | 5.7 [0.5] | 6.3 [0.3] | 4.9 [0.8] | 5.9 [0.3] | 4.8 [0.6] | 4.329 (3)† | 0.005 | 5.762 (1)†† | 0.017 | 1.101 (3)†† | 0.349 |
| % Achieving reversals | 100% | 100% | 98.6% | 94.4% | 100% | 87.5% | 96.9% | 94.7% | 2.26 (3)‡ | 0.598§ | 4.4 (1)‡ | 0.058§ | n/a | n/a |
| Win-switch rate, block-1 (90-50-10) | 0.09 [0.03] | 0.09 [0.04] | 0.07 [0.01] | 0.11 [0.05] | 0.04 [0.01] | 0.13 [0.05] | 0.1 [0.03] | 0.21 [0.06] | 2.284 (3)† | 0.079 | 7.117 (1)†† | 0.008 | 1.15 (3)†† | 0.329 |
| Win-switch rate, block-2 (80-40-20) | 0.05 [0.02] | 0.08 [0.03] | 0.04 [0.01] | 0.05 [0.04] | 0.02 [0.01] | 0.12 [0.05] | 0.06 [0.02] | 0.15 [0.05] | 2.067 (3)† | 0.105 | 9.918 (1)†† | 0.002 | 1.174 (3)†† | 0.32 |

*Table 2 continued on next page*

*Table 2 continued*

| | Version 1 | | Version 2 | | Version 3 | | Version 4 | | Version Effect | | Paranoia Effect | | Interaction | |
|---|---|---|---|---|---|---|---|---|---|---|---|---|---|---|
| | Low Paranoia (n=45) | High Paranoia (n=20) | Low Paranoia (n=69) | High Paranoia (n=18) | Low Paranoia (n=56) | High Paranoia (n=16) | Low Paranoia (n=64) | High Paranoia (n=19) | Statistic | p-value | Statistic | p-value | Statistic | p-value |
| Lose-stay rate, block 1 (90-50-10) | 0.27 [0.03] | 0.34 [0.05] | 0.37 [0.03] | 0.34 [0.04] | 0.3 [0.03] | 0.39 [0.06] | 0.32 [0.03] | 0.34 [0.04] | 0.561 (3)[†] | 0.641 | 1.834 (1)[††] | 0.177 | 0.754 (3)[††] | 0.521 |
| Lose-stay rate, block 2 (80-40-20) | 0.28 [0.03] | 0.23 [0.05] | 0.4 [0.03] | 0.32 [0.05] | 0.33 [0.03] | 0.37 [0.06] | 0.29 [0.03] | 0.33 [0.06] | 2.47 (3)[†] | 0.062 | 0.177 (1)[††] | 0.674 | 0.834 (3)[††] | 0.476 |
| Reaction time, block 1 | 433.6 [28.8] | 789.3 [282.7] | 548.1 [77.8] | 365.6 [24.4] | 448 [60.1] | 442.1 [59.5] | 557.2 [108.2] | 530 [130.2] | 0.793 (3)[†] | 0.499 | 0.161 (1)[††] | 0.689 | 1.727 (3)[††] | 0.161 |
| Reaction time, block 2 | 370.7 [23.3] | 494.3 [88.6] | 465.3 [61.6] | 331.4 [22.9] | 391.7 [52.3] | 555.9 [121.2] | 385.4 [29.2] | 504.1 [82.7] | 0.394 (3)[†] | 0.757 | 1.92 (1)[††] | 0.167 | 1.949 (3)[††] | 0.122 |

† Univariate analysis, F(df) with df error = 306 Exact test, ‡chi-square coefficient (df), § Exact significance (2-sided), ‖ Monte Carlo significance (2-sided).

**Table 3.** ANOVA results for HGF parameters.

| | Block effect [†] | | Group effect[‡] | | Interaction effect | |
|---|---|---|---|---|---|---|
| | Statistic[§] | p-value | Statistic[§] | p-value | Statistic[§] | p-value |
| | | | Experiment 1 | | | |
| $\omega_3$ | 11.672 (1) | **0.002** | 1.294 (1) | 0.264 | 6.948 (1) | **0.013** |
| $\mu_3^0$ | 25.904 (1) | **1.809E-5** | 7.063 (1) | **0.012** | 5.344 (1) | **0.028** |
| $\kappa$ | 7.768 (1) | **0.009** | 7.599 (1) | **0.010** | 0.003 (1) | 0.960 |
| $\omega_2$ | 2.182 (1) | 0.150 | 4.186 (1) | **0.050** | 0.058 (1) | 0.811 |
| $\mu2^0$ | 4.831 (1) | **0.036** | 1.261 (1) | 0.270 | 0.370 (1) | 0.547 |
| BIC | 0.061 (1) | 0.807 | 8.801 (1) | **0.006** | 1.7 (1) | 0.202 |
| | | | Experiment 2, Version 3 | | | |
| $\omega_3$ | 14.932 (1) | **0.0002** | 1.128 (1) | 0.292 | 1.406 (1) | 0.240 |
| $\mu_3^0$ | 64.651 (1) | **1.54E-11** | 6.366 (1) | **0.014** | 0.003 (1) | 0.959 |
| $\kappa$ | 15.53 (1) | **0.0002** | 13.521 (1) | **0.0005** | 0.011 (1) | 0.916 |
| $\omega_2$ | 0.027 (1) | 0.869 | 8.70 (1) | **0.004** | 0.090 (1) | 0.765 |
| $\mu_2^0$ | 11.432 (1) | **0.001** | 0.030 (1) | 0.864 | 0.203 (1) | 0.653 |
| BIC | 1.110E-5 (1) | 0.997 | 16.336 (1) | **0.0001** | 1.678 (1) | 0.199 |
| | | | Experiment 3: Rats | | | |
| $\omega_3$ | 30.086 (1) | **6.2785E-5** | 4.579 (1) | **0.049** | 9.058 (1) | **0.009** |
| $\mu_3^0$ | 31.416 (1) | **5.0188E-5** | 8.454 (1) | **0.011** | 5.159 (1) | **0.038** |
| $\kappa$ | 9.132 (1) | **0.009** | 13.356 (1) | **0.002** | 2.644 (1) | 0.125 |
| $\omega_2$ | 32.192 (1) | **4.4173E-5** | 22.344 (1) | **0.0003** | 18.454 (1) | **0.001** |
| $\mu_2^0$ | 5.226 (1) | **0.037** | 0.368 (1) | 0.553 | 2.087 (1) | 0.169 |
| BIC | 5.052 (1) | **0.040** | 1.890 (1) | 0.189 | 0.331 (1) | 0.573 |

Block refers to first versus second half in human studies, Pre-Rx vs Post-Rx in rat studies.‡ Group refers to low versus high paranoia in humans, saline versus methamphetamine in rats §F-statistic (degrees of freedom); df error = 30 in Experiment 1, 70 in Experiment 2, Version 3, and 50 in Experiment 3: Rats; split-plot ANOVA (i.e., repeated measures with between-subjects factor).

Experiment 1, we observed an interaction between task block and paranoia group ($F$(1, 30)=5.344, p=0.028, $\eta_p^2$=0.151; *Table 1*): $\mu_3^0$ differed between high and low paranoia in both blocks (block 1, $F$(1, 30)=4.232, p=0.048, $\eta_p^2$=0.124, MD = 0.658, CI=[0.005,1.312]; block 2, F(1, 30)=7.497, p=0.010, $\eta_p^2$=0.20, MD = 1.598, CI=[0.406, 2.789]), but only low paranoia subjects significantly updated their priors between block 1 and block 2 ($F$(1, 30)=39.841, p=5.85E-07, $\eta_p^2$=0.570, MD = 1.504, CI=[1.017, 1.99]). In Experiment 2, the analogous task design (version 3) demonstrated significant effects of block ($F$(1, 70)=64.652, p=1.54E-11, $\eta_p^2$=0.480, MD = 1.303, CI=[0.980,1.627]) and paranoia ($F$(1, 70)=6.366, p=0.014, $\eta_p^2$=0.083, MD = 0.909, CI=[0.191, 1.628]; *Table 1*). Rats showed a similar effect following methamphetamine exposure with a significant time (Pre-Rx, Post-Rx) by treatment (methamphetamine, saline) interaction ($F$(1, 15)=5.159, p=0.038, $\eta_p^2$=0.256; pre versus post methamphetamine effect: $F$(1, 15)=12.186, p=0.003, MD = 1.265, CI=[−0.493, 2.037]; Pre-Rx mean [standard error]=−1.25 [0.56] saline, −0.77 [0.80] methamphetamine; Post-Rx: $m$ = −0.69 [0.74] saline, 0.58 [0.73] methamphetamine). Random effects meta-analyses confirmed significant cross-experiment replication of elevated $\mu_3^0$ in human participants with paranoia (in laboratory and online version 3; $MD_{META}$ = 1.110, CI=[0.927, 1.292], $z_{META}$ = 11.929, p=8.356E-33) and across humans with paranoia and rats exposed to methamphetamine ($MD_{META}$ = 2.090, CI=[0.123, 4.056], $z_{META}$ = 2.083, p=0.037). Both paranoid humans and rats administered chronic methamphetamine had strong beliefs that the task contingencies would change rapidly and unpredictably – in other words, they expected frequent reversal events. Methamphetamine exposure made rats behave like humans with high paranoia (*Figure 2b*, Post-Rx condition, orange). This is particularly striking when compared to human data from the first task block (before contingency transition), when task designs are most similar across experiments.

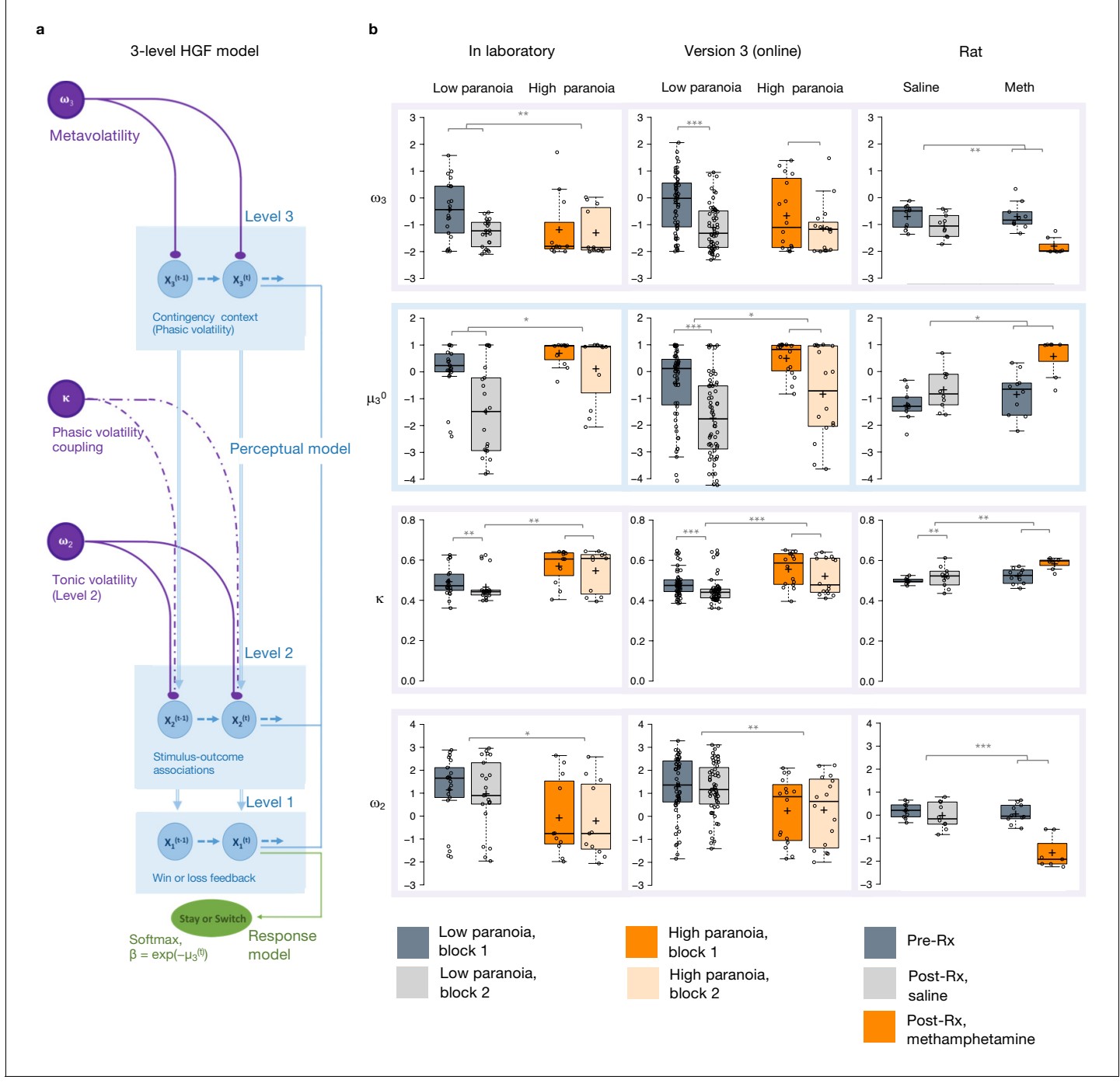

**Figure 2.** Hierarchical Gaussian Filter (HGF) model parameters. (a) 3-level HGF perceptual model (blue) with a softmax decision model (green). Level 1 ($x_1$): trial-by-trial perception of win or loss feedback. Level 2 ($x_2$): stimulus-outcome associations (i.e., deck values). Level 3 ($x_3$): perception of the overall reward contingency context. The impact of phasic volatility upon $x_2$ is captured by κ (i.e., coupling). Tonic volatility modulates $x_3$ and $x_2$ via $\omega_3$ and $\omega_2$, respectively. $\mu_3^0$ is the initial value of the third level volatility belief. (b) HGF model parameter estimates from each of our three studies (in laboratory, online, rat - columns), $\omega_3$, $\mu_3^0$, κ, and $\omega_2$, displayed hierarchically, in rows, in parallel with the position of the particular parameter in the model depiction in a). Parameters replicate across high paranoia groups in the in-laboratory experiment ($n$ = 21 low paranoia [gray], 11 high paranoia [orange]; dark bars are initial task blocks, lighter bars follow the contingency transition); the analogous online task (version 3, $n$ = 56 low paranoia [gray], 16 high paranoia [orange]; dark bars are initial task blocks, lighter bars follow the contingency transition); and rats exposed to chronic, escalating saline or methamphetamine ($n$ = 10 per group, Pre-Rx [dark gray]; Post-Rx, $n$ = 10 saline [light gray], seven methamphetamine [orange]). Center lines depict medians; box limits indicate the 25th and 75th percentiles; whiskers extend 1.5 times the interquartile range from the 25th and 75th percentiles, outliers are represented by dots; crosses represent sample means; data points are plotted as open circles. *p≤0.05, **p≤0.01, ***p≤0.001.

**Table 4.** Corrections for multiple comparisons.

| | Group effect [†] | | | | Interaction effect[‡] | | | |
|---|---|---|---|---|---|---|---|---|
| | Survives bonferroni?[§] | Survives FDR? | Critical value | Benjamini-Hochberg p-value | Survives bonferroni?[§] | Survives FDR? | Critical value | Benjamini-Hochberg p-value |
| Experiment 1 | | | | | | | | |
| $\omega_3$ | N/A | N/A | 0.05 | 0.264 | No | No | 0.0125 | 0.052 |
| $\mu_3^0$ | Yes | Yes | 0.025 | 0.024 | No | No | 0.025 | 0.056 |
| $\kappa$ | Yes | Yes | 0.0125 | 0.04 | N/A | N/A | 0.05 | 0.96 |
| $\omega_2$ | No | No | 0.0375 | 0.0667 | N/A | N/A | 0.0375 | 1.081 |
| Experiment 2, Version 3 | | | | | | | | |
| $\omega_3$ | N/A | N/A | 0.05 | 0.292 | N/A | N/A | 0.0125 | 0.96 |
| $\mu_3^0$ | No | Yes | 3.75E-02 | 0.0187 | N/A | N/A | 0.05 | 0.959 |
| $\kappa$ | Yes | Yes | 0.0125 | 0.002 | N/A | N/A | 0.0375 | 1.221 |
| $\omega_2$ | Yes | Yes | 0.025 | 0.008 | N/A | N/A | 0.025 | 1.53 |
| Experiment 3: Rats | | | | | | | | |
| $\omega_3$ | No | Yes | 5.00E-02 | 0.049 | Yes | Yes | 0.025 | 0.018 |
| $\mu_3^0$ | Yes | Yes | 3.75E-02 | 0.0147 | No | No | 0.0375 | 0.0507 |
| $\kappa$ | Yes | Yes | 0.025 | 0.004 | N/A | N/A | 0.05 | 0.125 |
| $\omega_2$ | Yes | Yes | 0.0125 | 0.0012 | Yes | Yes | 0.0125 | 0.004 |

N/A denotes to p-values that were not significant before corrections. † Low versus high paranoia in humans, saline versus methamphetamine in rats. ‡ Group by time (i.e., first versus second half in human studies, Pre-Rx vs Post-Rx in rat studies). § p-value < 0.0125.

Paranoid participants and methamphetamine exposed rats updated stimulus-outcome associations more strongly in response to perceived volatility (e.g., correctly or incorrectly inferred reversals; *Figure 2b*). $\kappa$ showed significant paranoia group and block effects across the in laboratory experiment and online version 3 (*Table 1*; paranoia effects, in laboratory: $F(1, 30)=7.599$, p=0.010, $\eta_p^2=0.202$, MD = 0.081, CI=[0.021, 0.140]; online version 3: $F(1, 70)=13.521$, p=0.0005, $\eta_p^2=0.162$, MD = 0.068, CI=[0.031–0.104]; $MD_{META} = 0.079$, CI=[0.063, 0.095], $z_{META} = 9.502$ p=2.067E-21); see *Table 3* for block effects). $\kappa$ increased from baseline in rats on methamphetamine, yielding significant effects of treatment ($F(1, 15)=13.356$, p=0.002, $\eta_p^2=0.471$, MD = 0.045, CI=[0.019, 0.072]) and time ($F(1, 15)=9.132$, p=0.009, $\eta_p^2=0.378$, MD = 0.041, CI=[0.012, 0.069]); however, the interaction between time and treatment did not reach statistical significance (*Table 3*; Pre-Rx $m = 0.499$ [0.015] saline, 0.523 [0.040] methamphetamine; Post-Rx: $m = 0.518$ [0.053] saline, 0.585 [0.029] methamphetamine). Replication of group effects was significant across all three experiments ($MD_{META} = 2.063$, CI=[0.341, 3.785], $z_{META} = 2.348$, p=0.019). Thus, learning was more strongly driven by unexpected uncertainty in high paranoia participants and rats chronically administered methamphetamine; they were faster to interpret volatility as reversal events than their low paranoia and saline exposed counterparts.

Expected uncertainty ($\omega_2$) was decreased in paranoid participants and rats exposed to methamphetamine (*Figure 2b*). In laboratory and online (version 3), paranoid individuals were slower to update stimulus-outcome associations in response to expected uncertainty (*Table 1*; $\omega_2$ paranoia effect, in laboratory: $F(1, 30)=4.186$, p=0.050, $\eta_p^2=0.122$, MD = −1.188, CI=[−2.375,–0.002]; online version 3: $F(1, 70)=8.7$, p=0.004, $\eta_p^2=0.111$, MD = −0.993, CI=[−1.665,–0.322]; $MD_{META} = -1.154$, CI=[−1.455,–0.853], $z_{META} = -7.521$, p=5.450E-14). The effects of methamphetamine exposure in rats were consistent ($MD_{META} = -1.992$, CI=[−3.318,–0.665], $z_{META} = -2.943$, p=0.003) yet more striking, with a strongly negative $\omega_2$ accounting for the more pronounced lose-stay behavior or perseveration in rats (time by treatment interaction, $F(1, 15)=18.454$, p=0.001, $\eta_p^2=0.552$; pre versus post methamphetamine: $F(1, 15)=42.242$, p=1.0E-5$^{22}$, $\eta_p^2=0.738$, MD = −1.604, CI=[−2.130,–1.078]; Pre-Rx $m = 0.198$ [0.33] saline, −0.036 [0.42] methamphetamine; Post-Rx: $m = -0.023$ [0.56] saline, −1.640 [0.71] methamphetamine). High paranoia humans and rats exposed to methamphetamine

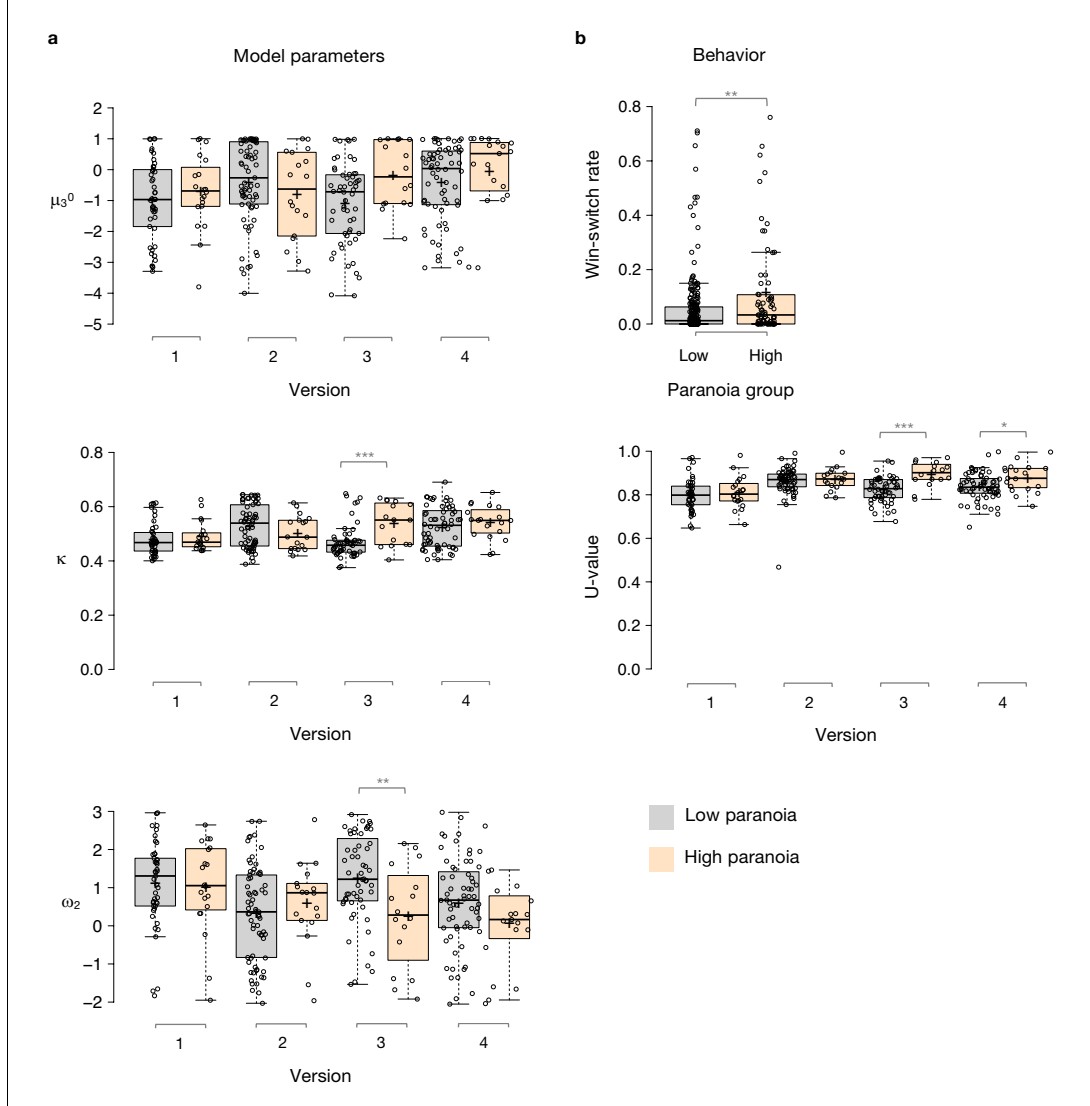

**Figure 3.** Paranoia effects across task versions. (a) Estimated model parameters derived from participant choices in response to the tasks. Low paranoia is shown in gray, high paranoia is shown in orange. $\mu_3^0$, $\kappa$, and $\omega_2$ are shown in separate panels (top, middle, and bottom panels, respectively; y-axes). X-axes depict each separate online task version from Experiment 2 (version 1: Easy-Easy, version 2: Hard-Hard, version 3: Easy-Hard, version 4: Hard-Easy). (b) Behavior. Win-switch rate (top): paranoid participants switched between decks more frequently after positive feedback. Rates are collapsed across all task versions and blocks (paranoia group effect; n = 234 low paranoia [gray], 73 high paranoia [orange]). U-value (bottom): a measure of choice stochasticity, calculated for low (gray) and high (orange) paranoia participants and collapsed across task blocks. U-values are shown separately for each online task version (1 through 4, as in part a). In versions 3 and 4 only (the versions containing unsignaled contingency transitions), paranoid participants showed higher U-values, suggesting increasingly stochastic switching rather than perseverative returns to a previously rewarding option. Center lines show the medians; box limits indicate the 25th and 75th percentiles; whiskers extend 1.5 times the interquartile range from the 25th and 75th percentiles, outliers are represented by dots; crosses represent sample means; data points are plotted as open circles. *P*-values correspond to estimated marginal means post-hoc comparisons: *p≤0.05, **p≤0.01, ***p≤0.001.

maintained rigid beliefs about the underlying option probabilities relative to low paranoia and saline controls. This was associated with perseverative behavior in the rats but not in humans.

Meta-volatility learning ($\omega_3$) was similarly decreased across paranoia and methamphetamine exposed groups (in laboratory, online version 3, and rats: $MD_{META} = -1.155$, CI=[−2.139,–0.171], $z_{META} = -2.3$, p=0.021), suggesting more reliance on expected task volatility (i.e., anticipated frequency of reversal events) than on actual task feedback. In laboratory, we observed a block by paranoia group interaction (*Table 1*, $F(1, 30)=6.948$, p=0.010, $\eta_P^2=0.188$). Post-hoc tests differentiated first and second blocks for the low paranoia group only ($F(1, 30)=26.640$, p=1.5E-5, $\eta_P^2=0.470$,

**Table 5.** Experiment 2 effects across block, paranoia group, and task version.

| | Block | | Group | | Version | | Block*group*Version | | Group*version | | Block*group | | Block*version | |
|---|---|---|---|---|---|---|---|---|---|---|---|---|---|---|
| | F (df)† | P | F (df)† | P | F (df)† | P | F (df)† | P | F (df)† | P | F (df)† | P | F (df)† | P |
| $\omega_3$ | 3.722 (1) | 0.055 | 0.499 (1) | 0.481 | 2.061 (3) | 0.105 | 0.415 (3) | 0.742 | 1.005 (3) | 0.391 | 0.145 (1) | 0.704 | 7.0155 (3) | 1.42E-4 |
| $\mu_3^0$ | 288.1 (1) | 1.01E-45 | 2.604 (1) | 0.108 | 2.321 (3) | 0.075 | 0.261 (3) | 0.853 | 2.329 (3) | 0.075 | 0.281 (1) | 0.597 | 0.061 (3) | 0.98 |
| $\kappa$ | 120.9 (1) | 7.65E-24 | 3.602 (1) | 0.059 | 5.06 (3) | 0.002 | 0.08 (3) | 0.971 | 4.178 (3) | 0.006 | 1.028 (1) | 0.312 | 2.559 (3) | 0.055 |
| $\omega_2$ | 35.3 (1) | 7.92E-9 | 4.435 (1) | 0.036 | 4.155 (3) | 0.007 | 0.166 (3) | 0.919 | 2.809 (3) | 0.04 | 2.387 (1) | 0.123 | 8.697 (3) | 1.5E-5 |
| $\mu_2^0$ | 71.3 (1) | 1.33E-15 | 0.242 (1) | 0.623 | 0.616 (3) | 0.605 | 1.081 (3) | 0.358 | 0.412 (3) | 0.744 | 0.057 (1) | 0.812 | 1.505 (3) | 0.213 |
| BIC | 56.6 (1) | 6.23E-13 | 8.073 (1) | 0.005 | 5.385 (3) | 0.001 | 0.262 (3) | 0.853 | 4.927 (3) | 0.002 | 0.451 (1) | 0.502 | 11.905 (3) | 2.19E-07 |

† F-statistic (degrees of freedom); df error = 299; split-plot ANOVA (i.e., repeated measures with two between-subjects factors).

N/A denotes to p-values that were not significant before corrections. † Low versus high paranoia in humans, saline versus methamphetamine in rats. ‡ Group by time (i.e., first versus second half in human studies, Pre-Rx vs Post-Rx in rat studies). § p-value < 0.0125.

MD = −0.876, CI=[−1.222,–0.529]). The paranoia effect did not reach statistical significance for online version 3 (block effect only, $F$(1, 70)=14.932, p=0.0002, $\eta_p^2$=0.176, MD = −0.692, CI= [−1.050,–0.335]; *Table 3*), but meta-analytic random effects analysis confirms a significant paranoia group difference (in laboratory and online version 3: $MD_{META}$ = −0.341, CI=[−0.522,–0.159], $z_{META}$ = −3.68, p=0.0002). Methamphetamine exposure rendered $\omega_3$ more negative in rats (time by treatment interaction, ($F$(1, 15)=9.058, p=0.009, $\eta_p^2$=0.376; pre versus post methamphetamine: $F$(1, 15)=30.668, p=5.7E-5, $\eta_p^2$=0.672, MD = −1.210, CI=[−1.676,–0.745]; Pre-Rx m = −0.692 [0.44] saline, −0.607 [0.51] methamphetamine; Post-Rx: $m$ = −1.044 [0.44] saline, −1.817 [0.32] methamphetamine). These data indicate that paranoia and methamphetamine are associated with slower learning about changes in task volatility, suggesting greater reliance on volatility priors than task feedback.

In summary, our modeling analyses suggest the following about paranoia in humans and methamphetamine exposed animals: they expect the task to be volatile (high $\mu_3^0$), their expectations about task volatility are more rigid (low $\omega_3$), and they confuse probabilistic errors and task volatility as a signal that the task has fundamentally changed (high $\kappa$, low $\omega_2$).

We applied False Discovery Rate (FDR) correction for multiple comparisons of each model parameter (*Hochberg and Benjamini, 1990*). $\kappa$ group effects survived corrections within each experiment (*Table 4*). In addition to $\kappa$, $\mu_3^0$ survived for experiment 1; $\mu_3^0$ and $\omega_2$ survived in online version 3; and $\mu_3^0$, $\omega_2$, and $\omega_3$ survived in experiment three as group effects. Such correction is not yet standard practice with this modeling approach (*Lawson et al., 2017*; *Powers et al., 2017*; *Sevgi et al., 2016*) but we believe it should be, and when effects survive correction we should increase our confidence in them.

## Paranoia effects across task versions

To examine the relationship between beliefs about contingency transition and paranoia within our HGF parameters, we performed split-plot, repeated measures ANOVAs across all four task versions. Paranoia group effects were specific to versions of the task in which we explicitly manipulated uncertainty via contingency transition which increased volatility (*Figure 3*, *Table 5*, versions 3 and 4). Specifically, we observed paranoia by version interactions for $\kappa$ ($F$(3, 299)=4.178, p=0.006, $\eta_p^2$=0.040) and $\omega_2$ ($F$(3, 299)=2.809, p=0.040, $\eta_p^2$=0.027; *Table 2*). Post-hoc tests confirmed that significant paranoia group effects were restricted to version 3 ($\kappa$: $F$(1, 299)=12.230, p=0.001, $\eta_p^2$=0.039, MD = 0.068, CI=[0.03,0.106]; $\omega_2$: $F$(1, 299)=8.734, p=0.003, $\eta_p^2$=0.028, MD = −0.993, CI=[−1.655,– 0.332]) and a trend for version 4 ($\omega_2$: $F$(1, 299)=2.909, p=0.089, $\eta_p^2$=0.010, MD = −0.528, CI= [−1.138, 0.081], *Figure 3a*). $\mu_3^0$ also exhibited a paranoia by version trend (*Table 2*, $F$(3, 299)=2.329, p=0.075, $\eta_p^2$=0.023), largely driven by version 3 ($F$(1, 299)=6.206, p=0.013, $\eta_p^2$=0.020, MD = 0.909, CI=[0.191, 1.628]; *Figure 3a*). There were no significant paranoia effects or interactions for $\omega_3$ (*Table 5*). In sum, our contingency shift manipulation – from easily discerned options to underlying

**Table 6.** Experiment 2 ANCOVAs.

| Effect | Df | ω3 F | ω3 p-value | μ30 F | μ30 p-value | κ F | κ p-value | ω2 F | ω2 p-value |
|---|---|---|---|---|---|---|---|---|---|
| **Demographics (age, gender, ethnicity, and race)** | | | | | | | | | |
| Block | 1, 294 | 0.328 | 0.568 | 10.835 | **0.001** | 3.425 | 0.066 | 2.711 | 0.101 |
| Block * Age | 1, 294 | 0.659 | 0.418 | 2.035 | 0.155 | 2.195 | 0.14 | 0.212 | 0.646 |
| Block * Gender | 1, 294 | 0.363 | 0.547 | 0.105 | 0.746 | 4.042 | **0.046** | 0.096 | 0.757 |
| Block * Ethnicity | 1, 294 | 0.016 | 0.901 | 0.042 | 0.837 | 0.268 | 0.605 | 0.024 | 0.876 |
| Block * Race | 1, 294 | 3.244 | 0.073 | 0.279 | 0.598 | 0.082 | 0.775 | 1.386 | 0.24 |
| Block * Paranoia Group | 1, 294 | 0.001 | 0.969 | 0.162 | 0.687 | 0.738 | 0.391 | 1.189 | 0.277 |
| Block * Version | 3, 294 | 7.61 | **7.25E-05** | 0.561 | 0.641 | 2.568 | 0.055 | 8.613 | **1.97E-05** |
| Block * Paranoia Group * Version | 3, 294 | 0.451 | 0.717 | 0.135 | 0.939 | 0.119 | 0.949 | 0.1 | 0.96 |
| Age | 1, 294 | 3.054 | 0.082 | 2.974 | 0.086 | 2.101 | 0.149 | 2.339 | 0.128 |
| Gender | 1, 294 | 0.438 | 0.509 | 0.02 | 0.886 | 0.005 | 0.941 | 0.014 | 0.905 |
| Ethnicity | 1, 294 | 0.029 | 0.865 | 0.059 | 0.808 | 0.087 | 0.768 | 0.221 | 0.639 |
| Race | 1, 294 | 0.072 | 0.789 | 2.218 | 0.138 | 0.373 | 0.542 | 0.333 | 0.564 |
| Paranoia Group | 1, 294 | 4.71E-04 | 0.983 | 0.741 | 0.39 | 1.795 | 0.182 | 3.302 | 0.071 |
| Version | 3, 294 | 1.845 | 0.14 | 1.914 | 0.128 | 4.975 | **0.002** | 3.786 | **0.011** |
| Paranoia Group * Version | 3, 294 | 0.935 | 0.424 | 1.911 | 0.129 | 3.599 | **0.014** | 1.919 | 0.127 |
| **Mental health factors (medication usage, diagnostic category, BAI score, and BDI score)** | | | | | | | | | |
| Block | 1, 257 | 3.333 | 0.069 | 95.753 | **3.12E-19** | 25.498 | **8.78E-07** | 8.341 | **0.004** |
| Block * BAI | 1, 257 | 0.26 | 0.611 | 1.532 | 0.217 | 2.852 | 0.093 | 0.394 | 0.531 |
| Block * BDI | 1, 257 | 0.009 | 0.926 | 0.208 | 0.649 | 6.55 | **0.011** | 0.597 | 0.441 |
| Block * Medication Usage | 1, 257 | 0.027 | 0.87 | 1.288 | 0.258 | 0.691 | 0.407 | 0.871 | 0.352 |
| Block * Diagnostic Category | 1, 257 | 1.366 | 0.244 | 1.785 | 0.183 | 0.063 | 0.803 | 0.208 | 0.649 |
| Block * Paranoia Group | 1, 257 | 0.068 | 0.795 | 0.298 | 0.586 | 0.298 | 0.586 | 0.007 | 0.935 |
| Block * Version | 3, 257 | 5.872 | **0.001** | 0.531 | 0.662 | 0.906 | 0.439 | 6.16 | **0.0005** |
| Block * Paranoia Group * Version | 3, 257 | 1.024 | 0.383 | 0.869 | 0.458 | 0.266 | 0.85 | 0.095 | 0.963 |
| BAI | 1, 257 | 1.108 | 0.294 | 0.012 | 0.913 | 0.954 | 0.33 | 0.921 | 0.338 |
| BDI | 1, 257 | 0.037 | 0.848 | 0.574 | 0.449 | 1.343 | 0.248 | 2.372 | 0.125 |
| Medication Usage | 1, 257 | 0.327 | 0.568 | 0.058 | 0.81 | 0.002 | 0.966 | 0.467 | 0.495 |
| Diagnostic Category | 1, 257 | 4.252 | **0.04** | 0.004 | 0.949 | 1.443 | 0.231 | 1.743 | 0.188 |
| Paranoia Group | 1, 257 | 0.057 | 0.811 | 0.233 | 0.63 | 1.032 | 0.311 | 1.695 | 0.194 |
| Version | 3, 257 | 3.183 | **0.025** | 2.73 | **0.045** | 5.274 | **0.002** | 4.468 | **0.004** |
| Paranoia Group * Version | 3, 257 | 0.311 | 0.818 | 2.307 | 0.077 | 4.556 | **0.004** | 3.397 | **0.019** |
| **Global cognitive ability (educational attainment, income, and cognitive reflection)** | | | | | | | | | |
| Block | 1, 290 | 1.19E-04 | 0.991 | 51.264 | **7.60E-12** | 28.675 | **1.83E-07** | 18.388 | **2.51E-05** |
| Block * Education | 1, 290 | 0.603 | 0.438 | 0.001 | 0.975 | 0.033 | 0.856 | 0.258 | 0.612 |
| Block * Income | 1, 290 | 1.211 | 0.272 | 2.874 | 0.091 | 3.483 | 0.063 | 2.421 | 0.121 |
| Block * Cognitive Reflection | 1, 290 | 1.83 | 0.177 | 0.709 | 0.401 | 1.221 | 0.27 | 4.667 | **0.032** |
| Block * Paranoia Group | 1, 290 | 0.005 | 0.946 | 0.359 | 0.55 | 0.263 | 0.608 | 0.885 | 0.348 |
| Block * Version | 3, 290 | 8.861 | **1.27E-05** | 0.182 | 0.909 | 2.325 | 0.075 | 8.815 | **1.35E-05** |
| Block * Paranoia Group * Version | 3, 290 | 0.826 | 0.48 | 0.478 | 0.698 | 0.15 | 0.929 | 0.3 | 0.825 |
| Education | 1, 290 | 0.111 | 0.739 | 0.578 | 0.448 | 1.395 | 0.239 | 0.608 | 0.436 |
| Income | 1, 290 | 2.763 | 0.098 | 1.382 | 0.241 | 0.055 | 0.814 | 1.035 | 0.31 |
| Cognitive Reflection | 1, 290 | 0.164 | 0.686 | 12.807 | **0.0004** | 0.224 | 0.636 | 0.807 | 0.37 |
| Paranoia Group | 1, 290 | 0.069 | 0.793 | 0.555 | 0.457 | 2.477 | 0.117 | 4.715 | **0.031** |

*Table 6 continued on next page*

*Table 6 continued*

| Effect | Df | ω3 F | ω3 p-value | μ30 F | μ30 p-value | κ F | κ p-value | ω2 F | ω2 p-value |
|---|---|---|---|---|---|---|---|---|---|
| Version | 3, 290 | 2.104 | 0.1 | 2.55 | 0.056 | 5.53 | **0.001** | 3.799 | **0.011** |
| Paranoia Group * Version | 3, 290 | 1.288 | 0.279 | 2.568 | 0.055 | 4.469 | **0.004** | 2.793 | **0.041** |

probabilities that are closer together – increased unexpected uncertainty the most, particularly in highly paranoid participants, compared to the other task versions.

## Covariate analyses

We completed three ANCOVAs for each HGF parameter derived from Experiment 2: demographics (age, gender, ethnicity, and race); mental health factors (medication usage, diagnostic category, BAI score, and BDI score); and metrics and correlates of global cognitive ability (educational attainment, income, and cognitive reflection; *Tables 6* and *7*). For κ, our metric of unexpected uncertainty, the paranoia by version interaction remained robust across all three ANCOVAs (demographics: $F_{(3, 294)}$ =3.753, p=0.011, $\eta_p^2$=0.037; mental health: $F_{(3, 257)}$=4.417, p=0.005, $\eta_p^2$=0.049; cognitive: $F_{(3, 290)}$ =4.304, p=0.005 $\eta_p^2$=0.043). The paranoia by version trend of $\mu_3^0$ diminished with inclusion of demographic, mental health, and cognitive covariates (demographic: $F_{(3, 294)}$=1.997, p=0.119, $\eta_p^2$=0.020; mental health: $F_{(3, 257)}$=1.942, p=0.123, $\eta_p^2$=0.022; cognitive: $F_{(3, 290)}$=2.193, p=0.089, $\eta_p^2$=0.022). The paranoia by version interaction for $\omega_2$ was robust to mental health and cognitive factors ($F_{(3, 257)}$=3.617, p=0.014, $\eta_p^2$=0.041; $F_{(3, 290)}$=3.017, p=0.030, $\eta_p^2$=0.030). A paranoia group effect and paranoia by version trend remained with inclusion of demographics ($\omega_2$, paranoia effect: $F_{(1, 294)}$ =4.275, p=0.040, $\eta_p^2$=0.014; interaction: $F_{(3, 294)}$=2.507, p=0.059, $\eta_p^2$=0.025). Thus κ – participants' perception of **unexpected uncertainty** – was the only parameter whose main effect of paranoia (higher κ in high paranoia participants) and paranoia-by-version interaction (higher κ in high paranoia participants as a function of increasing unexpected volatility in version 3) survived covariation for demographic, mental health and cognitive covariates. We are most confident that high paranoia participants have higher **unexpected uncertainty** which drives their excessive updating of stimulus-outcome associations.

## Relationships between parameters and paranoia

We found a significant correlation between κ and paranoia scores (*Figure 4*). However, depression and anxiety were also related to κ, and indeed, paranoia and depression correlate with one another, in our data and in other studies (*Na et al., 2019*). In order to explore commonalities among the rating scales in the present data, we performed a principle components analysis (*Figure 5*), identifying three principle components. The first principle component (PC 1) explained 82.3% of the variance in the scales and loaded similarly on anxiety, depression, and paranoia. It correlated significantly with kappa (r = 0.272, p=0.021). Depression, anxiety and paranoia all contribute to PC1. We suggest that this finding is consistent with the idea that depression and anxiety represent contexts in which paranoia can flourish and likewise, harboring a paranoid stance toward the world can induce depression and anxiety.

**Table 7.** Modified Cognitive Reflection Questionnaire Items.

| Item | Prompt |
|---|---|
| 1 | A folder and a paper clip cost $1.10 in total. The folder costs $1.00 more than the paper clip. How much does the paper clip cost? |
| 2 | If it takes 5 clerks 5 min to review five applications, how long would it take 100 clerks to review 100 applications? |
| 3 | In a garden, there is a cluster of weeds. Every day, the cluster doubles in size. If it takes 48 days for the cluster to cover the entire garden, how long would it take for the cluster to cover half of the garden? |

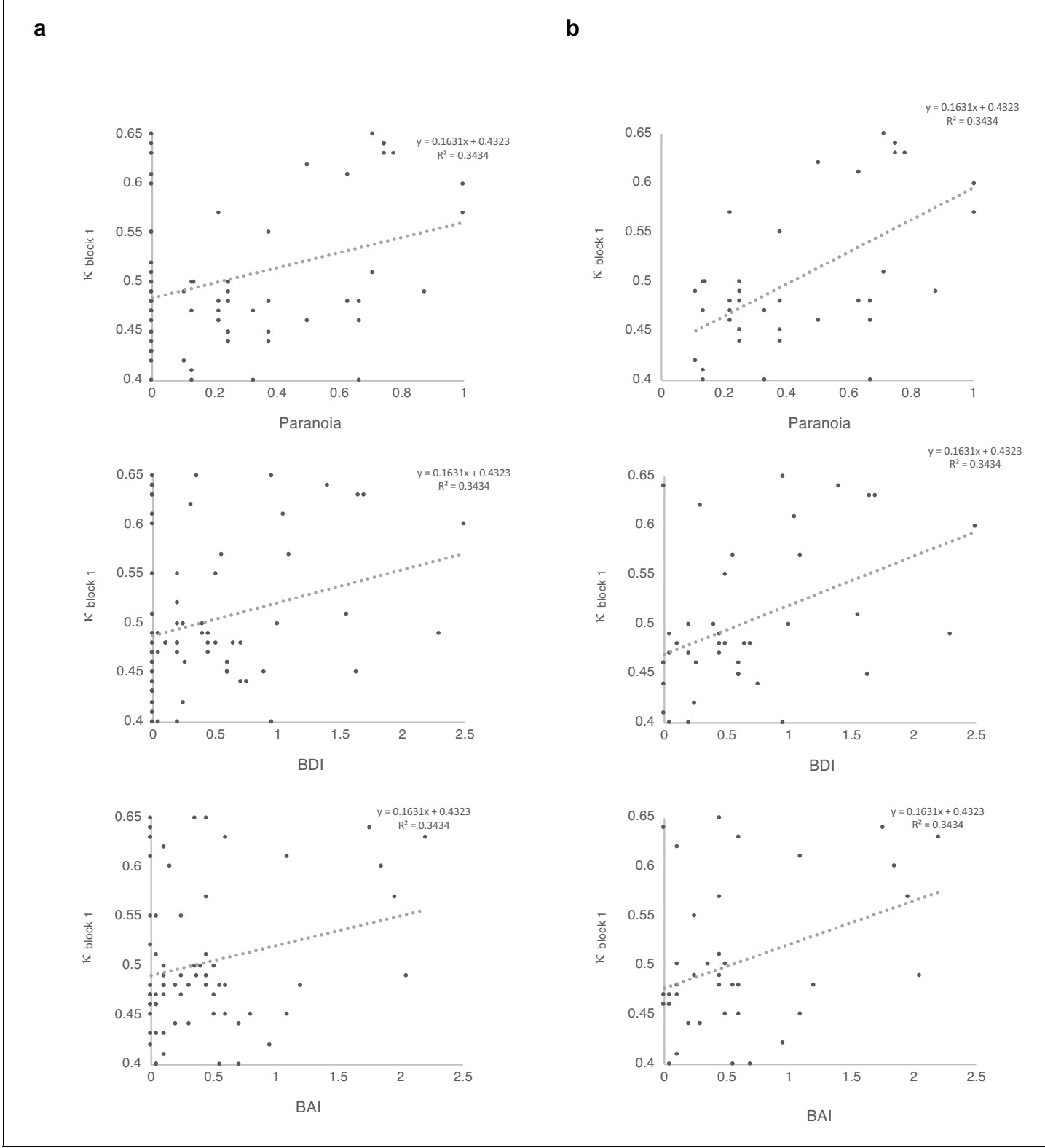

**Figure 4.** Correlations between κ and symptoms, with and without paranoia scores of zero. Paranoia (SCID-II, top), depression (BDI, middle), and anxiety (BAI, bottom). (**a**) Among all 72 subjects from online version 3, κ correlates with paranoia (r = 0.30, p=0.011, top) and depression (r = 0.250, p=0.034, middle), but not anxiety (r = 0.210, p=0.077, bottom). (**b**) Among participants who endorse at least one paranoia item (SCID-II paranoia >0, n = 39), κ correlates with paranoia (r = 0.588, p=8.1E-5, top), depression (r = 0.427, p=0.007, middle), and anxiety (r = 0.367, p=0.021, bottom). All correlations are two-tailed.

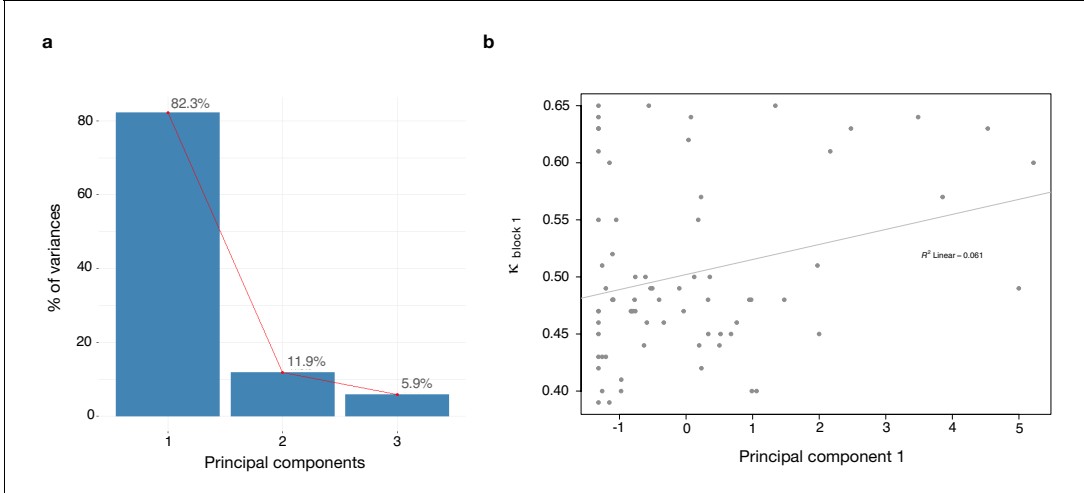

**Figure 5.** Dimensionality reduction analysis. Principal component analysis (PCA) was performed on behavioral data to explain the relationship between κ and the rating scales - paranoia (SCID), depression (BDI) and anxiety (BAI). (a) Scree plot of PCA illustrates percent of variance for each component explained by SCID, BDI and BAI. (b) Principal component 1 (PC1) plotted against κ values. κ correlates with PC1 (r = 0.272, p=0.021).

## Multiple regression

In order to make the case that our observations were most relevant to paranoia, we examined the effects of paranoia, anxiety, and depression on κ within the online version three dataset with multiple regression. A significant regression equation was found ($F_{(3,68)}$=3.681, p=0.016), with an R (*Freeman et al., 2005*) of 0.140. Participants' predicted κ equaled 0.486 + 0.062 (PARANOIA) +0.012 (BDI) −0.006 (BAI). Paranoia was a significant predictor of κ (β = 0.343, t = 2.470, p=0.016, CI=[0.012, 0.113]) but depression and anxiety were not (BDI: β = 0.086, t = 0.423, p=0.674, CI= [−0.043, 0.066]; BAI: β = −0.043, t = −0.218, p=0.828, CI=[−0.063, 0.050]). Examination of correlation plots for κ (*Figure 4*) revealed a much stronger relationship when analyses were restricted to individuals with paranoia scores greater than 0 (i.e., endorsement of at least one item); among participants who denied all questionnaire items, a minority (seven out of 32) exhibited elevated κ. To account for the possibility that some individuals with severe paranoia may avoid disclosing sensitive information, we performed additional analyses of participants who endorsed one or more paranoia item. The correlation between paranoia and κ in the first block of the task increases from r = 0.3, p=0.011, CI=[0.074, 0.497] (all participants, n = 72) to r = 0.588, p=8.0E-5, CI=[0.335, 0.762] (participants with paranoia >0, n = 39). In this subset, a significant regression equation was also found ($F_{(3,35)}$=6.322, p=0.002), with an $R^2$ of 0.351 (*Figure 4*). Participants' predicted κ was equal to 0.432 + 0.150 (PARANOIA)+0.013 (BDI) −0.004 (BAI). Paranoia was a significant predictor of κ (β = 0.538, t = 2.983, p=0.005, CI=[0.048, 0.252]) but depression and anxiety were not (BDI: β = 0.111, t = 0.494, p=0.624, CI=[−0.041, 0.067]; BAI: β = −0.035, t = −0.163, p=0.872, CI=[−0.057, 0.049]). Thus, paranoia predicts kappa across participants. Anxiety and depression do not predict kappa.

## Behavior and simulations

Win-switching was the prominent behavioral feature of both paranoid participants and rats exposed to methamphetamine (*Table 1*, *Table 2*; *Groman et al., 2018*). Collapsed across blocks and task versions, our Experiment 2 data demonstrated a main effect of paranoia group (*Figure 3b*; $F_{(1, 299)}$ =9.207, p=0.003, $\eta_p^2$=0.030, MD = 0.059, CI=[0.021, 0.097]; version trend: $F_{(3299)}$=2.263 p=0.081, $\eta_p^2$=0.022; low paranoia: $m$ = 0.06 [0.01], high paranoia: $m$ = 0.12 [0.02]). To elucidate whether this behavior was stochastic or predictable (e.g., switching back to a previously rewarding option), we calculated U-values (*Kong et al., 2017*), a metric of behavioral variability employed by behavioral ecologists (increasingly an inspiration for human behavioral analysis [*Fung et al., 2019*]), particularly with regards to predator-prey relationships (*Humphries and Driver, 1970*). When a predator is approaching a prey animal, the prey's best course of action is to behave randomly, or in a *protean* fashion, in order to evade capture (*Humphries and Driver, 1970*). The more protean or stochastic

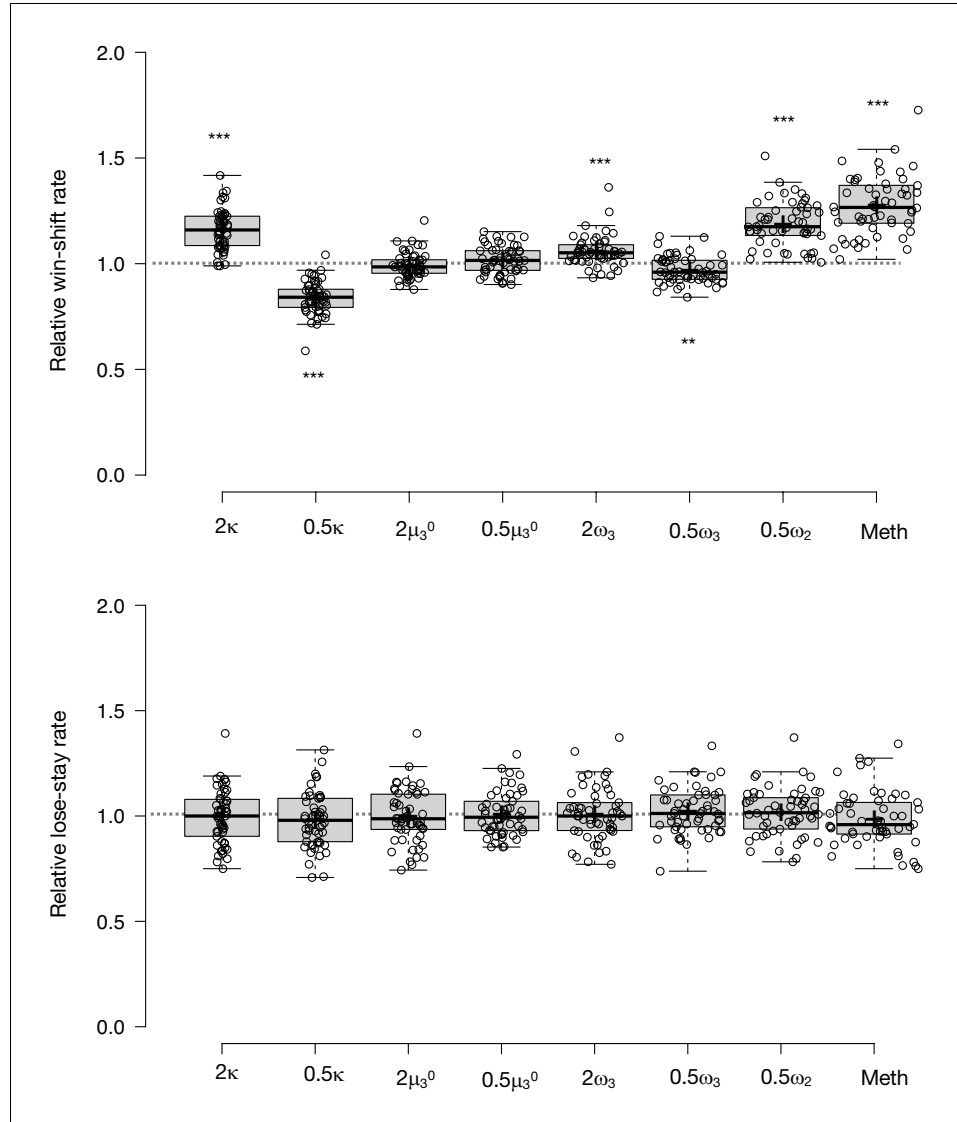

**Figure 6.** Parameter effects on simulated task performance. We simulated behavior from low paranoia participants (online Version 3, n = 54) to evaluate the effects of $\kappa$, $\mu_3^0$, $\omega_2$, and $\omega_3$ on win-shift and lose-stay rates. Estimated perceptual parameters were averaged across subjects to create a single set of baseline parameters. Additional parameter sets were created by doubling or halving one parameter at a time (e.g., 2 $\kappa$ or 0.5 $\kappa$), while the others were held constant (n.b., 2 $\omega_2$ violated model assumptions and was excluded from analysis). We also included the average parameter values of rats exposed to methamphetamine (Meth). Ten simulations were run per subject for each condition (i.e., parameter set). Win-shift and lose-stay rates were calculated, then averaged across simulations and subjects. Rates from each condition were divided by the baseline condition rate to generate relative win-shift and lose-stay rates. We compared relative rates for each condition to the baseline (relative rate of 1, depicted as the dotted line; paired t-tests, Bonferroni-corrected p-values). Of note, baseline parameters were positive for $\kappa$ and $\omega_2$, and negative for $\mu_3^0$ and $\omega_3$. Consequently, the doubled (2x) condition makes $\mu_3^0$ and $\omega_3$ more negative (lower). (n = 54). Box-plots: center lines show the medians; box limits indicate the 25th and 75th percentiles; whiskers extend 1.5 times the interquartile range from the 25th and 75th percentiles, outliers are represented by dots; crosses represent sample means; data points are plotted as open circles; *p≤0.05, **p≤0.01, ***p≤0.001.

the behavior, the closer to the U-value is to 1. Across task blocks, paranoid participants exhibited elevated choice stochasticity (paranoia by version interaction, $F(3, 298)=3.438$, p=0.017, $\eta_p^2=0.033$; *Table 2*). Post-hoc tests indicate that this stochasticity was specific to versions with contingency transition, suggesting a relationship to unexpected uncertainty (*Figure 3b*; version 3, $F(1, 298)=17.585$,

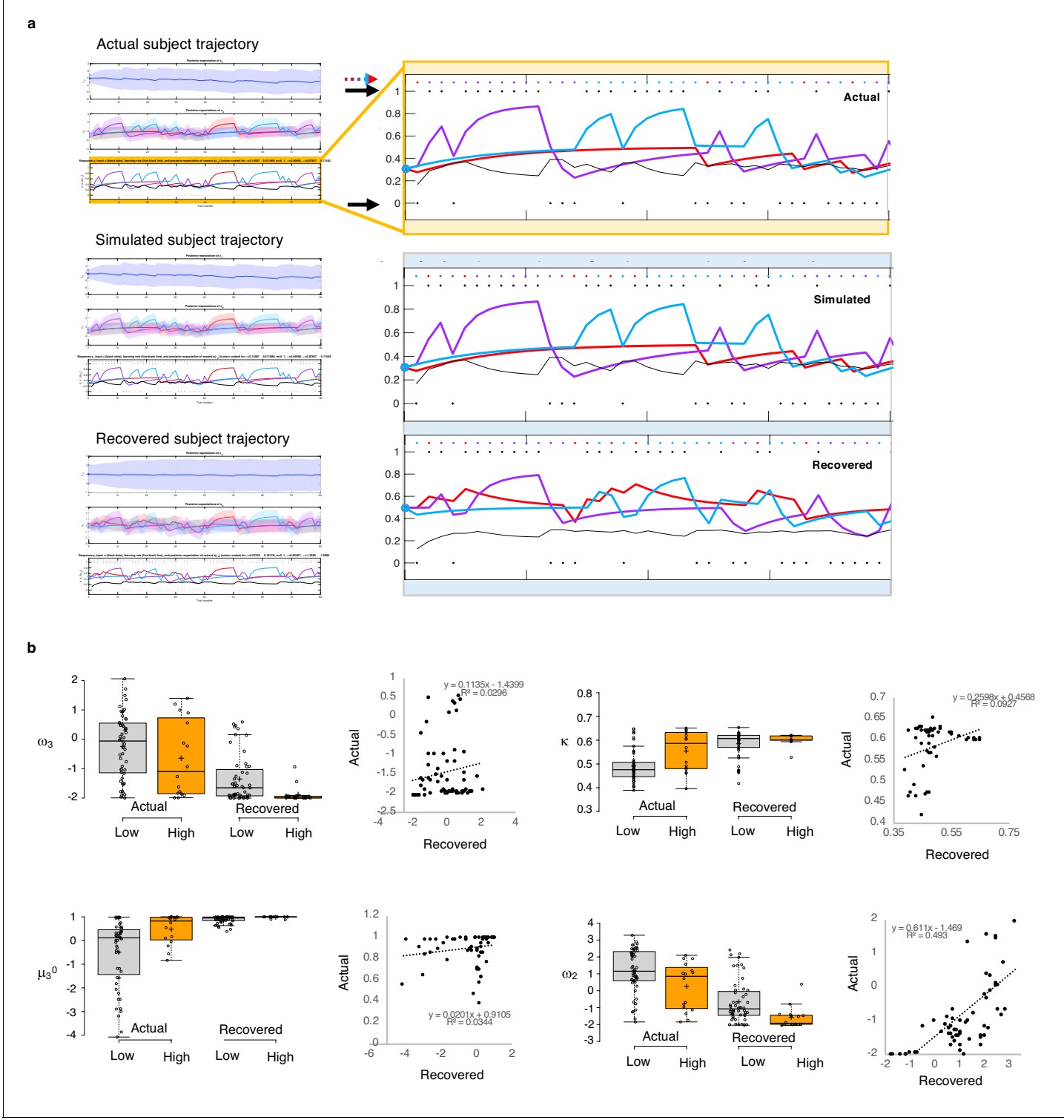

**Figure 7.** Parameter recovery. (**a**) Actual subject trajectory: this is an example choice trajectory from one participant (top). The layers correspond to the three layers of belief in the HGF model (depicted in *Figure 2a*). Focusing on the low-level beliefs (yellow box): The purple line represents the subject's estimated first-level belief about the value of choosing deck 1; blue, their belief about the value of choosing deck 2; and red, their belief about the value of choosing deck 3. Simulated subject trajectory represents the estimated beliefs from choices simulated from estimated perceptual parameters from that participant (middle), and Recovered subject trajectory represents what happens when we re-estimate beliefs from the simulated choices (bottom). Crucially, Simulated trajectories closely align with real trajectories (the increases and decreased in estimated beliefs about the values of each deck [purple, blue, red lines] align with each other across actual, simulated and recovered trajectories), although trial-by-trial choices (colored dots and

*Figure 7 continued on next page*

*Figure 7 continued*

arrow) occasionally differ. Outcomes (1 or 0; black dots and arrows) remain the same. (**b**) Actual versus Recovered: these data represent the belief parameters estimated from the participant's responses (Actual) compared to those estimated from the choices simulated with the participant's perceptual parameters (**Recovered**). Actual and Recovered values significantly correlate for $\omega_2$ (r = 0.702, p=2.52E-11) and $\kappa$ (r = 0.305, p=0.011) but not $\omega_3$ (r = 0.172, p=0.16) or $\mu_3^0$ (r = 0.186, p=0.13). Box plots: gray indicates low paranoia, orange designates high paranoia; center lines depict medians; box limits indicate the 25th and 75th percentiles; whiskers extend 1.5 times the interquartile range from the 25th and 75th percentiles, outliers are represented by dots; crosses represent sample means; data points are plotted as open circles. Online version three dataset.

p=3.6E-5, $\eta_p^2$=0.056, MD = 0.071, CI=[0.038, 0.104]; version 4, $F$(1, 298)=6.397, p=0.012, $\eta_p^2$=0.021, MD = 0.039, CI=[0.009, 0.07]). Our task manipulation, increasing unexpected volatility, increases win-switching behavior and stochastic choice more in more paranoid participants.

To test the propriety of our model, we simulated data for each subject in online version 3 and determined whether or not key behavioral effects (*Figure 7a*, *Table 1*, *Table 8*) were present. Using individually estimated HGF parameters to generate ten simulations per participant, we recapitulated both elevated win-switch behavior (paranoia effect, $F$(1, 70)=15.394, p=2.01E-4, $\eta_p^2$=0.180, MD = 0.186, CI=[0.091, 0.28]) and choice stochasticity (U-value; paranoia effect, F(1, 70)=13.362, p=0.0005, $\eta_p^2$=0.160, MD = 0.065, CI=[0.030, 0.101]) in simulated paranoid participants (*Figure 7b*; simulated win-switch rate, low paranoia: $m$ = 0.24 [0.02], high paranoia: $m$ = 0.43 [0.04]; simulated U-value, low paranoia: $m$ = 0.851 [0.008], high paranoia: $m$ = 0.916 [0.016]). Neither real nor simulated data showed any significant relationship between lose-stay behavior and paranoia (*Table 1*, *Table 2*, *Table 8*). To demonstrate the effects of parameters on task performance, we performed additional simulations in which we doubled or halved a single parameter at a time from the baseline average of low paranoia participants. These results confirmed the impact of $\kappa$, $\omega_2$, and $\omega_3$ on win-shift behavior (*Figure 4*). Parameter recovery revealed significant correlations for $\kappa$ and $\omega_2$ between original subject parameters and those estimated from simulations (*Figure 6*; $\omega$: r = 0.702, p=2.52E-11, CI=[0.557, 0.805]; $\kappa$: r = 0.305, p=0.011, CI=[0.072, 0.506]). Higher level parameters ($\omega_3$, $\mu_3^0$) were less consistently recovered, as noted in previous publications (*Bröker et al., 2018*). Thus, the model we chose, with meta-volatility and three coupled layers of belief, successfully simulates the key features of paranoid behavior: higher win-switching and stochastic choice.

## Alternate models

Our model is complex and other simpler reinforcement learning models might explain behavior on this task. Given the win-switching behavior we sought to understand, we fit a model from Lefebvre and colleagues that instantiated biased belief updating via differential weighting of positive and

**Table 8.** Simulations and behavior.

| Effect | Df | Win-switch rate F | Win-switch rate p-value | U-value F | U-value p-value | Lose-stay rate F | Lose-stay rate p-value |
|---|---|---|---|---|---|---|---|
| Experiment 1 | | | | | | | |
| Block | 1, 30 | 1.465 | 0.236 | 16.999 | **0.0003** | 1.334 | 0.257 |
| Block*Paranoia Group | 1, 30 | 0.602 | 0.444 | 2.393 | 0.132 | 2.575 | 0.119 |
| Paranoia Group | 1, 30 | 3.579 | 0.068 | 3.312 | 0.079 | 2.283 | 0.141 |
| Experiment 2, Version 3 | | | | | | | |
| Block | 1, 70 | 0.935 | 0.337 | 10.153 | **0.002** | 0.122 | 0.728 |
| Block*Paranoia Group | 1, 70 | 0.001 | 0.982 | 0.003 | 0.958 | 1.93 | 0.169 |
| Paranoia Group | 1, 70 | 12.698 | **0.001** | 19.209 | **4.03E-05** | 1.095 | 0.299 |
| Simulations[†] | | | | | | | |
| Block | 1, 70 | 0.176 | 0.676 | 3.335 | 0.072 | 5.073 | **0.027** |
| Block*Paranoia Group | 1, 70 | 2.039 | 0.158 | 2.624 | 0.11 | 0.036 | 0.85 |
| Paranoia Group | 1, 70 | 15.394 | **0.0002** | 13.362 | **0.0005** | 0.042 | 0.839 |

[†]Simulated data from experiment 2, Version 3.

**Table 9.** Alternative models fail to capture paranoia group differences.

| | Low Paranoia (n=56)† | | | High Paranoia (n=16)† | | | Paranoia Group Effect‡ | | Paranoia x Block Effect‡ | |
|---|---|---|---|---|---|---|---|---|---|---|
| | Mean | SEM | 95% CI | Mean | SEM | 95% CI | F(df) | P | F(df) | P |
| **Q-learning with learning rates for positive and negative prediction errors** | | | | | | | | | | |
| *Positive prediction error (α+)* | | | | | | | | | | |
| 1st half | 0.463 | 0.038 | [0.388, 0.538] | 0.475 | 0.071 | [0.335, 0.616] | 0.243 (1, 70) | 0.623 | 0.118 (1, 70) | 0.732 |
| 2nd half | 0.476 | 0.039 | [0.398, 0.555] | 0.535 | 0.074 | [0.379, 0.672] | | | | |
| *Negative prediction error (α-)* | | | | | | | | | | |
| 1st half | 0.421 | 0.022 | [0.377, 0.464] | 0.365 | 0.041 | [0.284, 0.446] | 1.292 (1, 70) | 0.260 | 0.320 (1, 70) | 0.573 |
| 2nd half | 0.386 | 0.021 | [0.344, 0.427] | 0.364 | 0.039 | [0.285,0.442] | | | | |
| *Inverse temperature (β )* | | | | | | | | | | |
| 1st half | 271 | 74.0 | [126, 416] | 147 | 133 | [-114, 408] | 1.626 (1, 70) | 0.207 | 0.043 (1, 70) | 0.837 |
| 2nd half | 316 | 82.3 | [155, 477] | 145 | 132 | [-114, 403] | | | | |
| **2-level HGF with softmax decision model** | | | | | | | | | | |
| *μ2* | | | | | | | | | | |
| 1st half | -0.059 | 0.081 | [-0.218, 0.100] | -0.303 | 0.157 | [-0.611, 0.005] | 3.039 (1, 70) | 0.086 | 0.385 (1, 70) | 0.537 |
| 2nd half | -0.244 | 0.082 | [-0.405, -0.082] | -0.566 | 0.155 | [-0.869, -0.262] | | | | |
| *Inverse temperature (β)* | | | | | | | | | | |
| 1st half | 131 | 30.6 | [71.3, 191] | 35.3 | 6.20 | [23.2, 47.5] | 2.665 (1, 70) | 0.107 | 0.250 (1, 70) | 0.619 |
| 2nd half | 119 | 30.6 | [58.7, 179] | 52.1 | 12.1 | [28.3, 75.9] | | | | |

† Online version 3 data ‡ Repeated measures ANOVA.

negative prediction errors (*Lefebvre et al., 2018*). Fitting this model to online version 3, we saw no significant paranoia group differences in learning rates for positive or negative prediction errors in parameters derived from all 180 trials (independent samples t-test: $α^+$, $t(70)=-0.532$, $p=0.597$; $α^-$, $t(70)=0.963$, $p=0.339$), nor did we see any significant block*paranoia or paranoia group effects by repeated measures ANOVA (block*paranoia: $α^+$, $F(1, 70)=0.188$, $p=0.732$, $α^-$, $F(1, 70)=0.378$, $p=0.540$; paranoia group: $α^+$, $F(1, 70)=0.243$, $p=0.623$, $α^-$, $F(1, 70)=1.292$, $p=0.260$). See *Table 9*.

We can also simplify within our hierarchical Gaussian Filter framework. The model we chose had three layers of beliefs and the highest level seemed to capture most of the task and paranoia effects of interest (*Figure 8*). To confirm this suspicion, we removed the third layer, fitting an HGF model that had beliefs about outcomes and deck values but no beliefs about volatility, no unexpected volatility learning rate, nor meta-volatility. This model failed to capture the task effects or group differences in its parameters (see *Table 9*).

Therefore, a more complicated model, one that captures higher-level beliefs about contingency transitions or learning when to learn, seems most appropriate, and indeed, that type of model was able to simulate the key features of our data (*Palminteri et al., 2017*). Future work will compare and contrast different potential computational models included, but not limited to Bayesian Hidden State Markov Models (*Hampton et al., 2006*), as well as switching (*Gershman et al., 2014*) and volatile Kalman Filters (*Piray and Daw, 2020*).

## Clustering analysis

Given the apparent similarity in effects of paranoia and methamphetamine in humans and rats, respectively (*Figure 2b*), we searched for latent structure in our data using two-step cluster analysis (*Tkaczynski, 2017*). This approach sorts subjects into groups (clusters) on the basis of some experimenter-selected variables such as estimated model parameters. The goal is to find distinct subsets in the data such that each cluster exhibits a cohesive pattern of relationships between the variables. Whereas some clustering approaches require the experimenter to predefine the expected number of clusters, two step-clustering determines both the optimal number of clusters and the composition

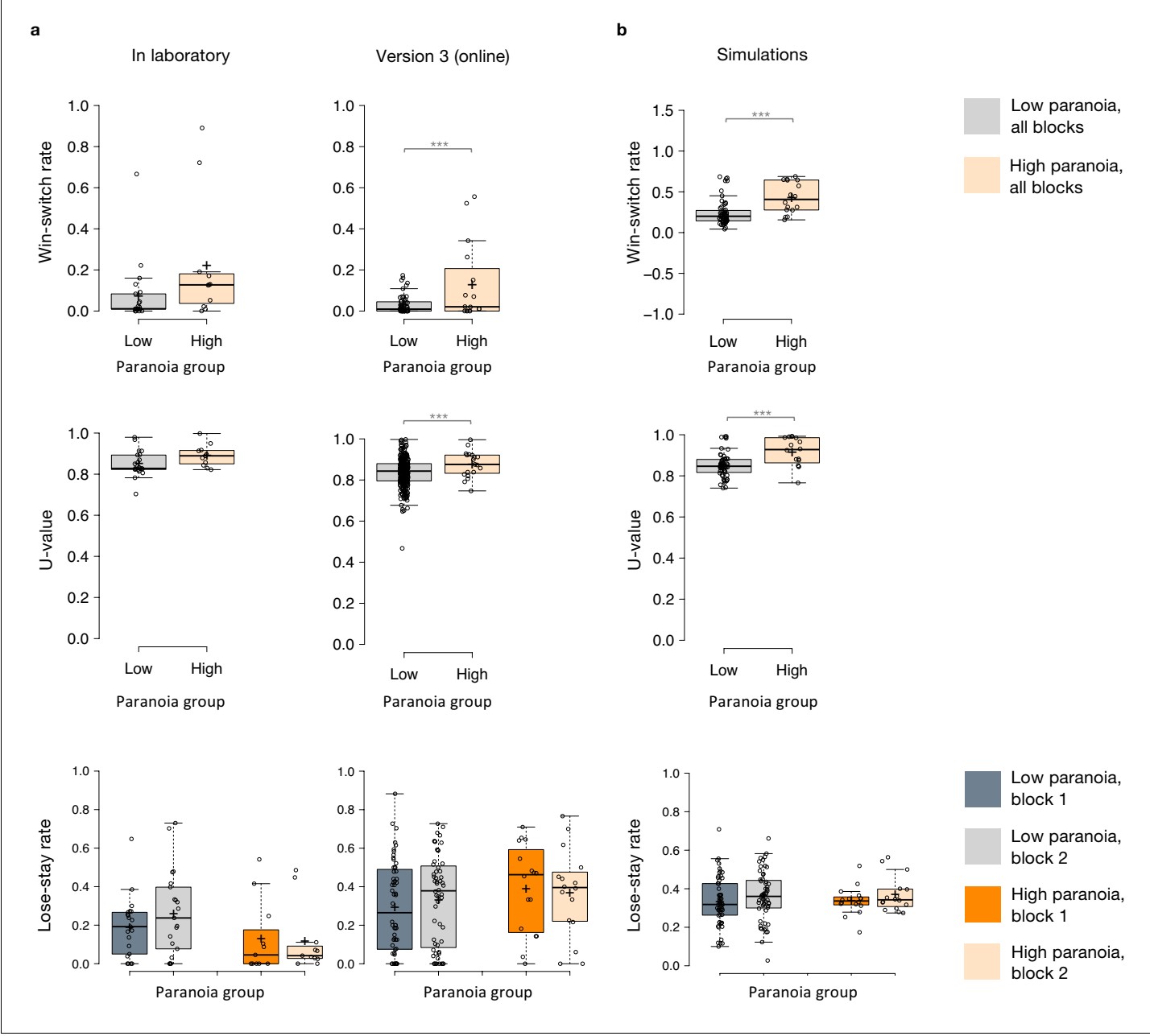

**Figure 8.** Behavioral data and simulations. (**a**) Plots of in laboratory and online behavioral metrics. Win-switch rate (switching after positive feedback), U-value (behavioral stochasticity) and Lose-stay rate (perseverating after a loss). Low paranoia participants are shown in gray, High paranoia in orange. Win-switch rates and U-values are collapsed across blocks. For Lose-stay rates, darker colors are block one data and lighter colors are block two data. Behavioral switching patterns replicate across in laboratory and online version three experiments. Perseveration after negative feedback (lose-stay behavior) did not significantly differ between paranoia groups or task block. (**b**) Simulated data generated from HGF perceptual parameters (version 3). Win-switch rate, U-value and Lose-stay rate of the simulated data are depicted. The model simulated data replicate the win-switch and U-value behavioral differences between high and low paranoia participants presented in panel **a**. Like the real participants, there was no difference in lose-stay rates in the simulated data. Center lines show the medians; box limits indicate the 25th and 75th percentiles; whiskers extend 1.5 times the interquartile range from the 25th and 75th percentiles, outliers are represented by dots; crosses represent sample means; data points are plotted as open circles. *p≤0.05, **p≤0.01, ***p≤0.001. Plots of participant behavioral metrics (**a**) are presented side by side with simulated data (**b**).

of each cluster. The greater the similarity (or homogeneity) within a group and the greater the difference between groups, the better the clustering.

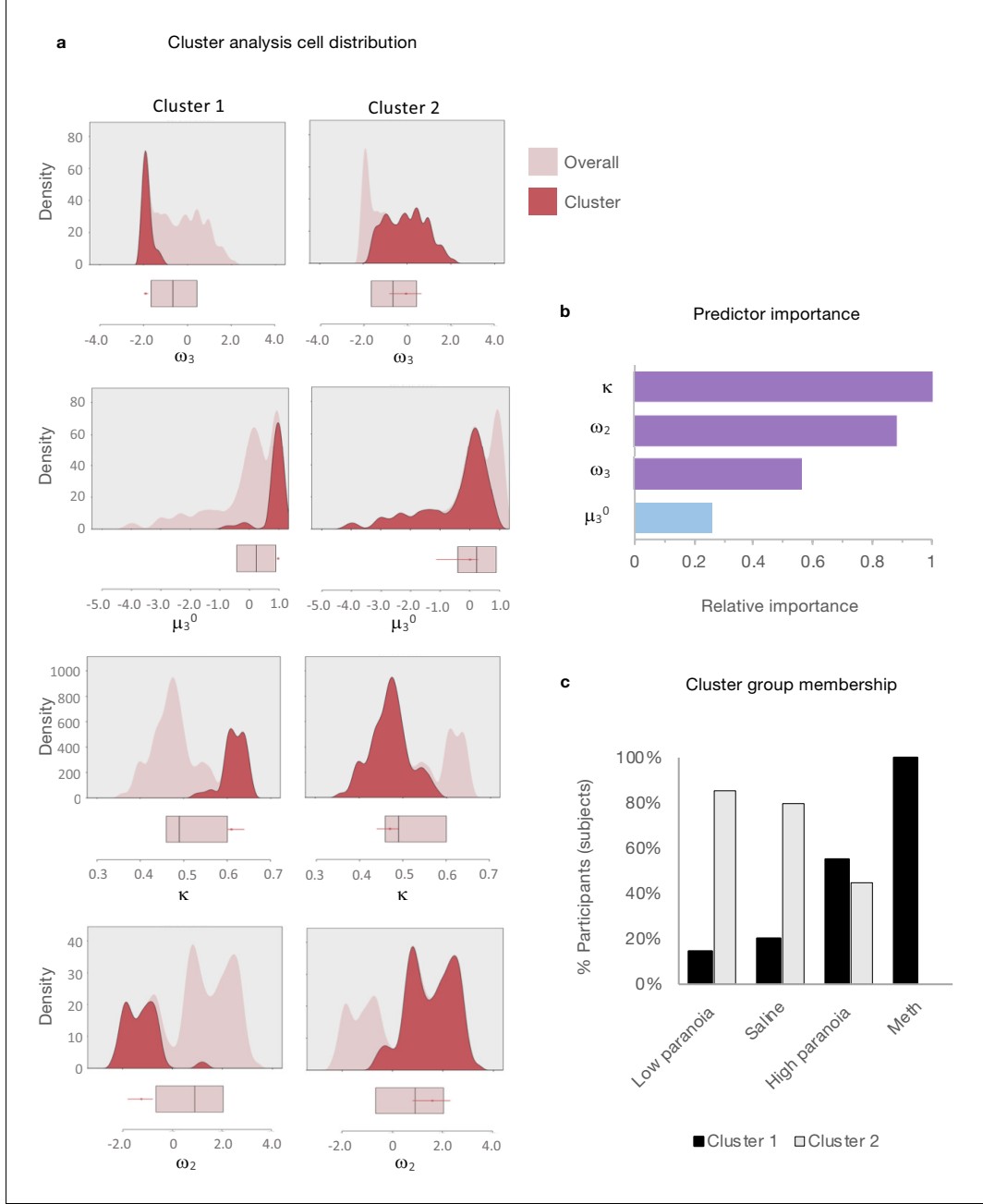

**Figure 9.** Cluster analysis of HGF parameters. Two-step cluster analysis of model parameters ($\omega_3$, $\mu_3^0$, $\kappa$, $\omega_2$) across rat and human data sets (rat, post-Rx; in laboratory and online version 3, block 1). Automated clustering yielded an optimal two clusters with good cohesion and separation (average silhouette coefficient = 0.7; cluster size ratio = 2.46). (a) **Density plots** for $\mu_3^0$, $\kappa$, $\omega_2$, and $\omega_3$ (light pink) depict cluster-specific distributions for each parameter (red). Unlike frequency histograms (that depict the number of data points in bins), density plots employ smoothing to prioritize distribution shape and are not restricted by bin size. Beneath each density plot, box-plots of overall median, 25th quartile, and 75th quartile for each parameter are aligned (pink), with cluster medians and quartiles superimposed (red). Relative to the overall distribution, Cluster 1 (*n* = 35) medians are elevated for $\mu_3^0$ and $\kappa$, decreased for $\omega_2$ and $\omega_3$. Cluster 2 (*n* = 86) falls within each overall distribution. (b) **Predictor importance** of included parameters. Consistent with the color scheme in **Figure 2a**, Uncertainty weighting parameters ($\kappa$, $\omega_2$, $\omega_3$ ) are depicted in purple and $\mu_3^0$ the prior is in blue. (c) **Distribution of cluster identities within groups**. Black bars signify the proportion of group members assigned to Cluster one and gray bars represent the proportion of group members assigned to Cluster 2. Cluster one membership is significantly associated with paranoia and methamphetamine groups ($\chi^2$(1, *n* = 121)=29.447, p=5.75E-8). Columns display means [standard error] or

*Figure 9 continued on next page*

*Figure 9 continued*

percentage of participants within the described category, test-statistics, and p-values. [†]Independent samples t-test: t-value (df). Two-tailed *P*-values reported. [‡]Chi square coefficient (df). [§]Fisher's exact test, exact significance (2-sided). [¶]Equal variances not assumed. [#]Not significant (Bonferonni correction). [††]Data presented in *Figure 8*; repeated measures ANOVA, paranoia group trend or effect: *F*(df), *P*; estimated marginal means and standard error. [‡‡]Data presented in *Figure 2*; repeated measures ANOVA, *F*(df), *P*. In laboratory: paranoia x block interactions for $\omega_3$, $\mu_3^0$; paranoia group effects for $\kappa$, $\omega_2$. Version 3: paranoia group effects reported. See *Table 3* for complete ANOVA. results. Version columns display means [standard error] or percentage of participants within the described category. [††]Univariate analysis, F(df). [‡]Exact test, chi-square coefficient (df). [§] Exact significance (2-sided). [‖]Monte Carlo significance (2-sided). [‡‡]Data presented in *Figure 3*; repeated measures ANOVA, *F*(df), *P*. Mean values collapsed across blocks.

---

Considering that paranoia and methamphetamine exposure share a pattern of elevated $\mu_3^0$ and $\kappa$ accompanied by decreased $\omega_2$ and $\omega_3$ (*Table 10*), we hypothesized that these four variables would yield a distinct cluster: a 'paranoid style' across species. We analyzed $\mu_3^0$, $\kappa$, $\omega_2$, and $\omega_3$ estimates derived from the first block of experiment one and online version 3 (pre-context change data, because rats do not experience a context shift) with post-chronic exposure rat data (methamphetamine and saline). We identified two clusters with good cohesion and separation, meaning that subjects sorted into two groups (each containing rodents and humans) whose parameters travelled in such a way that their values were close to the centroid or mean of the cluster they were in and as far as possible from the centroid of the other cluster (average silhouette coefficient = 0.7; cluster size ratio = 2.46; *Figure 9a*). All parameters contributed to clustering; $\kappa$ contributed most strongly (*Figure 9b*). Importantly, the cluster solution did not separate rats from humans (despite the differences in task structure, incentives, manipulanda, and phylogeny). Relative to the overall distribution, Cluster one was characterized by high $\kappa$ and $\mu_3^0$, and decreased $\omega_2$ and $\omega_3$. Cluster one membership was significantly associated with high paranoia and methamphetamine exposure, $\chi^2$(1, *n* = 121) =29.447, p=5.75E-8, Cramer's V = 0.493 (*Figure 9c*). Notably, no participants in the low paranoia group with paranoia scores above zero were ascribed Cluster one membership. The cluster solution was robust to validation by split-half analysis (removing half of the participants and repeating the clustering), removal of the rat subjects, and removal of human participants. In each case, we identified two clusters with good cohesion and separation (**Split-half 1**, n = 19 cluster 1, 42 cluster 2: silhouette coefficient = 0.6; **Split-half 2**, n = 17 cluster 1, 43 cluster 2: silhouette coefficient = 0.7; **No Rat**, n = 26 cluster 1, 78 cluster 2: silhouette coefficient = 0.7; **Rat Only**, n = 6 cluster 1, 11 cluster 2: silhouette coefficient = 0.7). In summary, paranoid participants and methamphetamine-exposed rats cluster together (high $\mu_3^0$, high $\kappa$, low $\omega_2$, and low $\omega_3$), suggesting that these parameters share an underlying generative process and that paranoia and methamphetamine have similar effects on reversal-learning.

**Table 10.** Summary of paranoia/methamphetamine effects on belief-updating.

| | In lab | Online | Rats |
|---|---|---|---|
| $\omega_3$ | ↓[†] | ⇊ | ↓ |
| $\mu_3^0$ | ⬆ | ⬆[‡§] | ⬆ |
| $\kappa$ | ⬆ | ⬆[‡] | ⬆ |
| $\omega_2$ | ⬇ | ⬇[‡¶] | ⬇ |
| $\mu_2^0$ | - | - | - |

↑ ↓ Non-significant increase/decrease in high paranoia or meth, relative to low paranoia or saline ⇈ ⇊ Trend-level increase/decrease in high paranoia or meth, relative to low paranoia or saline ⬆⬇ Significantly higher/lower in high paranoia or meth, relative to low paranoia or saline - - No significant findings or trends † Baseline trend; parameter decreases in second block for low but not high paranoia ‡ Version 3 only § Trend-level significance disappears with inclusion of demographic covariates ¶ Significance reduced to trend with inclusion of demographic covariates.

## Discussion

During non-social probabilistic reversal-learning, paranoid individuals and rats chronically exposed to methamphetamine have higher initial expectations of task volatility ($\mu_3^0$). In other words, they start the task anticipating more changes in stimulus-outcome associations, and they switch choices readily and excessively in anticipation of reversal events. By relying more on their expectations of volatility than on actual experience (exemplified by switching even after positive feedback), they are slower to learn about changes in task volatility. This manifests as decreased meta-volatility learning ($\omega_3$) and failure to significantly adjust $\mu_3^0$ after contingency transitions. More paranoid individuals are similarly slower to adjust expected deck values (lower $\omega_2$) but faster to attribute volatility to reversal events (elevated $\kappa$), perceiving change (unexpected uncertainty) instead of normal statistical variation (expected uncertainty). They sit at Hofstadter's 'turning point', constantly expecting change but never learning appropriately from it.

In the reversal learning literature, choice switching after positive feedback has garnered less attention than perseverative behavior and sensitivity to negative feedback (*Izquierdo et al., 2017*; *Waltz, 2017*). Individuals with depression and schizophrenia seemingly perseverate less than healthy controls, but this has formerly been attributed to increased sensitivity to negative feedback (*Waltz, 2017*; *Robinson et al., 2012*). However, elevated win-switch tendencies have been reported in youths with bipolar disorder, major depressive disorder, and anxiety disorder (*Dickstein et al., 2010*). A prior study in people with schizophrenia described excessive win-switch behavior that correlated with the severity of delusional beliefs and hallucinations (*Waltz, 2017*). Likewise, an elevated prior on environmental volatility ($\mu_3^0$) and higher sensitivity to this volatility ($\kappa$) have been observed in HGF analyses of 2-choice probabilistic reversal-learning in medicated and unmedicated patients with schizophrenia (*Deserno, 2018*). These authors did not explore paranoia specifically.

We assessed paranoia across the continuum of health and mental illness, provided three choice options, and explicitly manipulated unexpected volatility across task versions. The version that shifted from an easier to a more difficult contingency context (version 3) was associated with paranoia group effects on $\mu_3^0$, $\kappa$, and $\omega_2$, and a meta-analytic effect on $\omega_3$. Furthermore, this contingency transition – an exposure to truly unexpected volatility – rendered low paranoia controls more similar to their paranoid counterparts by decreasing their meta-volatility learning ($\omega_3$). Paranoid participants responded to contingency transitions in version 3 and version four by switching stochastically. These

**Table 11.** Questionnaire item completion (% responses).

| Questionnaire/subscale | Experiment 1 | Experiment 2 |
|---|---|---|
| Age | 90.6% | 99.7% |
| Gender | 100.0% | 100.0% |
| Ethnicity | 100.0% | 100.0% |
| Race | 100.0% | 100.0% |
| Education | 100.0% | 99.7% |
| Meds | 100.0% | 90.6% |
| Dx | 100.0% | 94.1% |
| Income | N/A | 98.0% |
| SCID-II Paranoia - all items | 96.9% | 94.1% |
| SCID-II Paranoia - one item missing | 3.1% | 5.5% |
| SCID-II Paranoia - three items missing | 0.0% | 0.3% |
| Cognitive reflection - all items | N/A | 97.7% |
| Beck's Anxiety Inventory (BAI) - all items | 90.6% | 96.7% |
| BAI - one item missing | 3.1% | 2.9% |
| BAI - two items missing | 6.3% | 0.3% |
| Beck's Depression Inventory (BDI) - all items | 100.0% | 99.0% |
| BDI - one item missing | 0.0% | 1.0% |

findings suggest a continuum of behavioral responses to volatility, moving from optimal learning to diminished feedback sensitivity (i.e, decreased $\omega_3$ in low paranoia participants) and from diminished feedback sensitivity (lower $\omega_3$ and increased win-switching in high paranoia participants) toward complete dissociation from experienced feedback (stochastic switching).

Unexpected uncertainty, the perception of change in the probabilities of the environment — particularly 'unsignaled context switches" (*Yu and Dayan, 2005*) which increase unexpected volatility — is thought to promote abandonment of old associations and new learning. However, our results suggest that this response might vary according to a hierarchy of belief. Paranoid participants were quick to abandon 'best deck' associations and explore alternative options (i.e., $x_2$ beliefs), but in turn they relied more on their higher-level beliefs about the task volatility ($x_3$ beliefs) and less on sensory feedback (lower metavolatility learning). Our analysis of covariates warrants specific focus on $\kappa$, the sensitivity to unexpected volatility. Other parameter-paranoia associations did not endure after controlling for demographic factors (age, gender, ethnicity, and race), although we see their derangement in our rodent study as well as in the significant meta-analytic effects across our experiments. Furthermore, these demographic factors are themselves strong predictors of paranoia (*Holt and Albert, 2006*; *Iacovino et al., 2014*; *Mahoney et al., 2010*). It is notable too that $\kappa$ was the most powerful discriminator of the two clusters of human and animal participants. We conclude that elevated $\kappa$ - belief updating tethered to unexpected volatility - is the parameter change most robustly associated with paranoia. Doubling $\kappa$ in our simulations induced significantly more win-switching.

Multiple neurobiological manipulations may induce such win-switching behavior. Lesions of the mediodorsal thalamus in non-human primates (*Chakraborty et al., 2016*) or neurons projecting from the amygdala to orbitofrontal cortex in rats (*Groman et al., 2019*) engender win-switching. Unexpected uncertainty, and the $\kappa$ parameter of the HGF in particular (*Marshall et al., 2016*), are thought to be signaled via the locus coeruleus and noradrenaline (*Yu and Dayan, 2005*; *Payzan-LeNestour and Bossaerts, 2011*; *Payzan-LeNestour et al., 2013*; *Tervo et al., 2014*). This mechanism is thought to modulate switching versus staying behaviors (*Kane et al., 2017*; *Aston-Jones and Cohen, 2005*; *Aston-Jones et al., 1999*; *Eldar et al., 2013*), as well as responses to stress (*Borodovitsyna et al., 2018*; *McCall et al., 2015*; *Atzori et al., 2016*) and subliminal fear cues (*Liddell et al., 2005*) to coordinate fight-or-flight responses (*Atzori et al., 2016*). The dual role of the locus coeruleus in recognizing and responding to threats as well as unexpected uncertainty suggests that dysfunction could produce both paranoia and the inferential abnormalities we observed. Methamphetamine may induce similar dysfunction (*Ferrucci et al., 2019*; *Ferrucci et al., 2013*; *Ferrucci et al., 2008*). Acute moderate doses increase pre-synaptic catecholamine release, particularly noradrenaline (*Rothman et al., 2001*), and induce exploratory locomotive effects modulated through adrenoceptors on dopamine neurons (*Ferrucci et al., 2013*).

Excessive release of noradrenaline from the locus coeruleus into the anterior cingulate cortex drives feedback insensitivity and stochastic switching behavior in rats completing a three-option counter prediction task (*Tervo et al., 2014*). Evolutionarily, departure from predictable, rational actions might offer an adaptive mechanism for escape from intractable threat. As a protean defense mechanism, behavioral stochasticity impedes predators' abilities to create accurate, actionable countermeasures (*Humphries and Driver, 1970*; *Richardson et al., 2018*; *Humphries and Driver, 1967*). If driven by excessive unexpected uncertainty, underwritten by noradrenaline, protean defense may represent a heavily conserved, continuous common mechanism underlying vigilance and false alarms (*Aston-Jones et al., 1994*; *Rajkowski et al., 1994*; *Usher et al., 1999*), arousal-linked attentional biases (*Eldar et al., 2013*) and selective processing of social threats. However, protean behaviors are not necessarily adaptive. Pathological insensitivity to feedback and reliance on internal beliefs over evidence constitute a 'break from reality' – in other words, psychosis.

Efference copy models of motor control *Wolpert and Ghahramani, 2000* have been evoked to explain psychotic symptoms (*Blakemore et al., 2000*; *Blakemore et al., 1998*; *Blakemore et al., 1999*; *Blakemore et al., 2002*; *Frith et al., 2000a*; *Frith et al., 2000b*; *Shergill et al., 2005*; *Shergill et al., 2014*). Aberrant mismatches between expected and experienced sensory consequences of actions, weighted by their uncertainty (*Wolpert and Ghahramani, 2000*), can lead to the misattribution of one's movements to an external agent (*Blakemore et al., 2000*; *Blakemore et al., 1998*; *Blakemore et al., 1999*; *Blakemore et al., 2002*; *Frith et al., 2000a*; *Frith et al., 2000b*; *Shergill et al., 2005*; *Shergill et al., 2014*). Since we model others' intentions with reference to our model of ourselves (*Friston and Frith, 2015*), volatile experiences of ones' body and actions will

lead to uncertain and ultimately more threatening inferences about others (*Friston and Frith, 2015*). This would be entirely consistent with the present observations.

When confronted with intractable unexpected uncertainty our participants rely on higher-level beliefs about the task environment. When humans experience non-social volatility, (For example through threats to their sense of control [*Whitson and Galinsky, 2008*] or exposure to surprising non-social stimuli [*Proulx et al., 2012*; *Heine et al., 2006*]), they appeal to the influence of powerful enemies, even when those enemies' influence is not obviously linked to the volatility (*Sullivan et al., 2010*). Our account places the locus of paranoia at the level of the individual. Here, our account departs from evolutionary accounts of paranoia grounded in coalitional threat (*Raihani and Bell, 2019*; persecutors are not scapegoats that increase group cohesion. Rather, when paranoid, we have a ready explanation for hazards. With a well-defined persecutor in mind, a volatile world may be perceived to have less randomly distributed risk (*Sullivan et al., 2010*). However, paranoia might become a self-fulfilling prophecy, engendering more volatility and negative social interactions. This aspect may be captured in our task through win-switch behavior. By failing to incorporate positive feedback from the best option, paranoid individuals sample sub-optimal options which delivers misleading positive feedback.

There are some important limitations to our conclusions. Compared with humans, rats are relatively asocial. But they are not completely asocial. In our experiment they were housed in pairs, and, more broadly, they evince social affiliative interactions with other rats (*Donaldson et al., 2018*; *Kondrakiewicz et al., 2019*; *Urbach et al., 2010*). A further limitation centers on the comparability of our experimental designs. In humans our comparisons were both within (contingency transition) and between groups (low versus high paranoia). In rats, the model was also mixed with some between (saline versus methamphetamine) and some within-subject (pre versus post chronic treatment) comparisons. We should be clear that there was no contingency context transition in the rat study. However, just as that transition made low paranoia humans behave like high paranoia, chronic methamphetamine exposure made rats behave on a stable contingency much like high paranoia humans - even in the absence of contingency transition. The comparable results across species, despite these differences, warrant the inference that our basic, relatively asocial, approach provides a robust tool for computational dissection of learning mechanisms.

Social interactions play a rich and undeniable role in paranoia, but translational, domain-general approaches may ultimately facilitate biological insights into paranoia, psychosis and delusions (*Corlett and Fletcher, 2014*; *Feeney et al., 2017*). Whilst we contend that our task is relatively free of social features (certainly compared to others [*Raihani and Bell, 2017*]), the possibility remains that the elevated U-values in our participants are reflective of attempts (and perhaps failures) to predict our intentions as experimenters. Indeed, this is a possibility raised previously with regards to simple conditioned behaviors in experimental animals. Even during Pavlovian conditioning, animals may attempt to infer a generative model of the task environment, which might, ultimately, include the experimenter arranging the contingencies (*Gershman and Niv, 2012*; *Gershman and Niv, 2010*). It is possible that all instances of human cognitive testing involve an element of inference by the participant with regards to the intentions of the experimenter, whether or not the task at hand is explicitly social, and indeed, all cognitive functions may be aimed at or modulated by such inferences (*Turner et al., 1994*).

In summary, a strong belief in the volatility of the world necessitates hypervigilance and a facility with change. However, in paranoia, that belief in the volatility of the world is itself resistant to change, making it difficult to reassure, teach, or change the minds of people who are paranoid. They remain 'on guard,' adhering to expectations over evidence. By using a non-social task, we have shown that this paranoid style is not restricted to the social domain, and that it can be modeled in relatively asocial animals. Additionally, our domain-general approach reaffirms the merit of establishing expectations of a stable, predictable environment to promote recovery from paranoia-associated illness (*Powers et al., 2018*). We note with interest the apparent relationship between conspiratorial ideation and societal crisis situations (terrorist attacks, plane crashes, natural disasters or war) throughout history, with peaks around the great fire of Rome (AD 64), the industrial revolution, the beginning of the cold war, 9/11, and contemporary financial crises (*van Prooijen and Douglas, 2017*). In today's world of escalating uncertainty and volatilty – particularly environmental climate change and viral pandemics – our findings suggest that the paranoid style of inference may prove particularly maladaptive for coordinating collaborative solutions.

## Materials and methods

Experiments were conducted at Yale University and the Connecticut Mental Health Center (New Haven, CT) in strict accordance with Yale University's Human Investigation Committee and Institutional Animal Care and Use Committee. Informed consent was provided by all research participants.

### Experiment 1

English-speaking participants aged 18 to 65 (*n* = 34) were recruited from the greater New Haven area through public fliers and mental health provider referrals. Exclusion criteria included history of cognitive or neurologic disorder (e.g., dementia), intellectual impairment, or epilepsy; current substance dependence or intoxication; cognition-impairing medications or doses (e.g. opiates, high dose benzodiazepines); history of special education; and color blindness. Participants were classified as healthy controls (*n* = 18), schizophrenia spectrum patients (schizophrenia or schizoaffective disorder; *n* = 8), and mood disorder patients (depression, bipolar disorder, generalized anxiety disorder, post-traumatic stress disorder; *n* = 8) on the basis of clinician referrals and/or self-report. Participants were compensated $10 for enrolment with an additional $10 upon completion. Two healthy controls were excluded from analyses due to failure to complete the questionnaires and suspected substance use, respectively.

### Experiment 2

332 participants were recruited online via Amazon Mechanical Turk (MTurk). The study advertisement was accessible to MTurk workers with a 90% or higher HIT approval rate located within the United States. To discourage bot submissions and verify human participation, we required participants to answer open-ended free response questions; submit unique, separate completion codes for the behavioral task and questionnaires; and enter MTurk IDs into specific boxes within the questionnaires. All submissions were reviewed for completion code accuracy, completeness of responses (i. e., declining no more than 30% of questionnaire items), quality of free response items (e.g., length, appropriate grammar and content), and use of virtual private servers (VPS) to submit multiple responses and/or conceal non-US locations (Dennis VPS paper, 2018). Upon approval, workers were compensated $6. Those who scored in the top 25% on the card game (reversal-learning task) earned a $2 bonus. We rejected or excluded 19 submissions that geolocation services (https://www.iplocation.net/) identified as originating outside of the United States or from suspected server farms, four submissions for failure to manually enter MTurk ID codes, and two submissions for insufficient questionnaire completion. Submissions with grossly incorrect completion codes were rejected without further review.

### Experiment 3

Subject information, behavioral data acquisition, and behavioral analyses were described previously (*Groman et al., 2018*). Long Evans rats (Charles River; *n* = 20) ranged from 7 to 9 weeks of age. Rats were exposed to escalating doses and frequency of saline (*n* = 10) or methamphetamine (*n* = 10, three withdrawn during dosing), imitating patterns of human methamphetamine users (*Segal et al., 2003*; *Han et al., 2011*). Prior to dosing (Pre-Rx), rats completed 26 within-session reversal sessions, including up to eight reversals per session. Post-dosing (Post-Rx), rats completed one test session per week for four weeks. Computational model parameters were estimated from each session and averaged across treatment conditions to yield one Pre-Rx and Post-Rx set of parameters per rat.

### Behavioral task

Participants completed a 3-option probabilistic reversal-learning paradigm. Three decks of cards were displayed on a computer monitor for 160 trials. Participants selected a deck on each trial by pressing the predesignated key. We advised participants that each deck contained winning and losing cards (+100 and −50 points), but in different amounts. We also stated that the best deck may change. Participants were instructed to find the best deck and earn as many points as possible. Probabilities switched between decks when the highest probability deck was selected in 9 out of 10 consecutive trials (performance-dependent reversal). Every 40 trials the participant was provided a break, following which probabilities automatically reassigned (performance-independent reversal).

In Experiment 1, the task was presented via Eprime 2.0 software (Psychology Software Tools, Sharpsburg, PA). Participants were limited to a 3 s response window, after which the trial would time out and record a null response. A fixation cross appeared during variable inter-trial intervals (jittering). Task pacing remained independent of response time. In block 1 (trials 1–80) the reward probabilities (contingency) of the three decks were 90%, 50%, and 10% (90-50–10%). Without cue or warning (i.e. unsignaled to the participants) the contingency transitioned to 80%, 40%, and 20% (80-40–20%) upon initiation of block 2 (trials 81–160).

In Experiment 2, the task was administered via web browser link from the MTurk marketplace. We changed the task timing to self-paced and eliminated null trials and inter-trial jittering. A progress tracker was provided every 40 trials. Workers were randomly assigned to one of four task versions, using restricted block randomization to ensure comparable numbers of high paranoia participants across task versions. Version one had a constant contingency of 90-50–10%. Version 4 maintained a constant contingency of 80-40–20%. Version 3 replicated the 90-50–10% (block 1) to 80-40–20% (block 2) context transition of Experiment 1. Version 4 presented the reversed contingency transition, 80-40–20% (block 1) to 90-50–10% (block 2). We analyzed attrition rates across the four versions.

## Questionnaires

Following task completion, questionnaires were administered via the Qualtrics survey platform (Qualtrics Labs, Inc, Provo, UT). Items included demographic information (age, gender, educational attainment, ethnicity, and race) and mental health questions (past or present diagnosis, medication use, *Structured Clinical Interview for DSM-IV Axis II Personality Disorders* (SCID-II) (*Ryder et al., 2007*), Beck's Anxiety Inventory (BAI) (*Beck et al., 1988*), Beck's Depression Inventory (BDI) (*Beck et al., 1961*). We removed the single suicidality question from the BDI for Experiment 2. Experiment 2 included additional items: income, three cognitive reflection questions (*Table 7*), and three free response items ('What do you think the card game was testing?', 'Did you use any particular strategy or strategies? If yes, please describe', and 'Did you find yourself switching strategies over the course of the game?'). We quantified trait-level paranoia using the paranoid personality subscale of the SCID-II, and we included an ideas of reference item from the schizotypy subscale ('When you are out in public and see people talking, do you often feel that they are talking about you?') This item, along with other SCID-II items, has previously been included as a metric of paranoia in the general population (*Bebbington et al., 2013*; *Bell and O'Driscoll, 2018*). Participants who endorsed four or more paranoid personality items (i.e., the cut-off for the top third identified in Experiment 1) were classified as 'high paranoia.' Each participant's SCID-II, BAI, and BDI scores were normalized by total scale items answered. Response rates were higher than 90% for all questionnaire items and scales (*Table 11*).

## Behavioral analysis

We analyzed tendencies to choose alternative decks after positive feedback (win-switch) and select the same deck after negative feedback (lose-stay). Win-switch rates were calculated as the number of trials in which the participant switched after positive feedback divided by the number of trials in which they received positive feedback. Lose-stay rates were calculated as number of trials in which a participant persisted after negative feedback divided by total negative feedback trials. In Experiment 1, we excluded post-null trials from these analyses. To further characterize switching behavior, we calculated U-values, a measure of choice stochasticity:

$$U-value = -\Sigma_{i=1}^{\beta} \frac{\log(\alpha_i) \, x \, \alpha_i}{\log(\beta)} \quad (1)$$

where $\beta$ is the number of possible choice options (i.e., card decks or noseports) and $\alpha$ equals the relative frequency of choice option $i$ (*Kong et al., 2017*). To avoid any choice counterbalancing effects across reversals, choice frequencies were determined by the underlying probabilities of the decks rather than their physical attributes (e.g., deck position or color). Additional behavioral analyses included trials to first reversal, trials to post-reversal recovery, and trials to post-reversal switch. The latter two were restricted to the first reversal in the first block. Trials post-reversal were counted

from the first-negative feedback trial following the true reversal event. Recovery was defined as switching to the best deck and staying for at least one additional trial.

## Computational modeling

### Materials

The Hierarchical Gaussian Filter (HGF) toolbox v5.3.1 is freely available for download in the TAPAS package at https://translationalneuromodeling.github.io/tapas (*Mathys et al., 2011*; *Mathys et al., 2014*). We installed and ran the package in MATLAB and Statistics Toolbox Release 2016a (Math-Works, Natick, MA).

### Perceptual parameter estimation

In the human reversal-learning experiments, we estimated perceptual parameters individually for the first and second halves of the task (i.e., blocks 1 and 2). Each participant's choices (i.e., deck 1, 2, or 3) and outcomes (win or loss) were entered as separate column vectors with rows corresponding to trials. Wins were encoded as '1', losses as '0', and choices as '1', '2', or '3'. We selected the autoregressive 3-level HGF multi-arm bandit configuration for our perceptual model and paired it with the softmax-mu03 decision model.

Rat reversal-learning data was entered similarly, with choices designated as '1', '2', or '3' and reward presence or absence noted as '1' and '0', respectively. Perceptual parameters were estimated as a single block per session and averaged across Pre-Rx or Post-Rx sessions for each subject. Since the contingency remained 70-30–10%, we used the default start point values of $\mu_2$ and $\mu_3$, as in block one estimations for the human reversal-learning experiments).

### Simulations

We performed ten simulations per participant (online version 3) to determine whether our parameter estimates and model successfully captured behavioral differences between groups (e.g., win-switch rates). Each simulation required the participant's actual data (i.e., the column vectors 'outcomes' and 'choices') and the corresponding set of derived perceptual parameters. On each trial, a new choice was simulated conditional on the actual inputs in previous trials.

To illustrate the effects of each parameter on task behavior we doubled or halved one parameter at a time, by establishing a baseline set of perceptual parameters containing the average values from the low paranoia participants (online version 3). We then ran 10 simulations per subject for each of the following conditions: baseline, $2\kappa$, $0.5\kappa$, $2\mu_3^0$, $0.5\mu_3^0$, $2\omega_3$, $0.5\omega_3$, $2\omega_2$, $0.5\omega_2$, and the average perceptual parameters ($\kappa$, $\mu_3^0$, $\omega_3$, and $\omega_2$) from Post-Rx methamphetamine rats. The $2\omega_2$ condition yielded parameters in a region where model assumptions were violated (negative posterior precision error message) and was excluded from further analysis. Win-shift and lose-stay rates were calculated from each simulation as follows, and then averaged for each condition:

$$Win-switch\ rate = \frac{Number\ of\ trials\ in\ which\ choice\ switched\ after\ positive\ feedback}{Total\ positive\ feedback\ trials}$$

$$Lose-stay\ rate = \frac{Number\ of\ trials\ in\ which\ choice\ repeated\ after\ negative\ feedback}{Total\ negative\ feedback\ trials}$$

For each participant, we divided rates derived from each condition by the baseline rates to determine relative win-switch and lose-stay rates. We compared each relative rate to the baseline condition (i.e., 1.0) with paired-samples t-tests using Bonferroni-corrected p-values.

### Parameter recovery

We performed perceptual parameter estimation (see above) on 10 simulations per subject using first block data from online version 3. These simulations were generated from each subject's corresponding perceptual parameters. We averaged recovered parameters across simulations and low versus high paranoia (*Figure 7*).

## Alternative models

We employed a Q-learning model with separate parameter weights for positive and negative prediction errors to determine whether differential weighting might contribute to paranoia group effects. This model has been described previously (*Lefebvre et al., 2018*). We also evaluated whether a simpler two-level HGF model might suffice to capture paranoia group differences. To sever the third level from the model, we fixed the log- $\kappa$ parameter at negative infinity (i.e., by additionally setting the variance to zero), and similarly fixed the values of $\mu_3$, $\omega_3$, $\omega_2$, $\Phi_3$ at the values previously assigned in the configuration file. Parameter estimation was performed as described above, with a softmax decision model.

## Statistics

Unless otherwise specified, statistical analyses and effect size calculations were performed in IBM SPSS Statistics, Version 25 (IBM Corp., Armonk, NY), with an alpha of 0.05. Box-plots were created with the web tool BoxPlotR (*Spitzer et al., 2014*). Model parameters were corrected for multiple comparisons using the Benjamini Hochberg (False Discovery Rate) method. Bonferroni corrected results were largely consistent (*Table 4*).

To compare questionnaire item means between two groups (*Table 1*, low versus high paranoia), we conducted independent samples t-tests. To compare questionnaire item means across paranoia groups and task versions (*Table 2*), we employed univariate analyses. Associations between characteristic frequencies and subject group or task version were evaluated by Chi-Square Exact tests (two groups) or Monte Carlo tests (more than two groups). Pearson correlations established the associations between paranoia and BDI scores, BAI scores, win-switch rates, and $\kappa$. We selected two-tailed p-values where applicable and assumed normality. Multiple regressions were conducted with $\kappa$ estimates from the first task block (dependent variable) and paranoia, BAI, and BDI scores from online version 3.

To compare HGF parameter estimates and behavioral patterns (win-switch, U-value, lose-stay) across block, paranoia group (Experiment 1, Experiment 2 version 3), and/or task version (Experiment 2), we employed repeated measures and split-plot ANOVAs (i.e., block designated within-subject factor, paranoia group and task version as between subject). We similarly evaluated Experiment three parameter estimates for treatment by time interactions. For Experiment 2, we performed ANCOVAs for $\mu_3^0$, $\kappa$, $\omega_2$, and $\omega_3$ to evaluate three sets of covariates: (1) demographics (age, gender, ethnicity, and race); (2) mental health factors (medication usage, diagnostic category, BAI score, and BDI score); (3) and metrics and correlates of global cognitive function (educational attainment, income, and cognitive reflection). Unless otherwise stated, post-hoc tests were conducted as least significant difference (LSD)-corrected estimated marginal means.

Meta-analyses were conducted using random effects models with the R Metafor package (*Viechtbauer, 2010*). Mean differences were assessed for low versus high paranoia groups in the in-laboratory experiment and online version 3. Standardized mean differences (methamphetamine or high paranoia versus saline or low paranoia) were employed to account for the differences in task design between animal and human studies.

The 2-step clustering analysis approach was selected to automatically determine optimal cluster count and cluster group assignment. Clustering variables included paranoia-relevant parameter estimates ($\mu_3^0$, $\kappa$, $\omega_2$, and $\omega_3$) from Experiment 1 (block 1); online, version 3 (block 1), and rats (Post-Rx) as continuous variables with a Log-likelihood distance measure, maximum cluster count of 15, and Schwarz's Bayesian Criterion (BIC) clustering criterion. We validated our clustering solution by sorting the data into two halves and running separate cluster analyses. We also compared cluster solutions derived exclusively from rat data versus human data. A Chi-Square test determined the significance of the association between cluster membership and group (methamphetamine/high paranoia versus saline/low paranoia).

## Data availability

Data are available on ModelDB (*McDougal et al., 2017*; http://modeldb.yale.edu/258631) with accession code p2c8q74m.

## Acknowledgements

This work was supported by the Yale University Department of Psychiatry, the Connecticut Mental Health Center (CMHC) and Connecticut State Department of Mental Health and Addiction Services (DMHAS). It was funded by an IMHRO/Janssen Rising Star Translational Research Award, an Interacting Minds Center (Aarhus) Pilot Project Award, NIMH R01MH12887 (PRC), NIMH R21MH120799-01 (PRC and SG). EJR was supported by the NIH Medical Scientist Training Program Training Grant, GM007205; NINDS Neurobiology of Cortical Systems Grant, T32 NS007224; and a Gustavus and Louise Pfeiffer Research Foundation Fellowship. SU received funding from NSF Fellowships DGE1122492 and DGE1752134. SMG and JRT were supported by NIDA DA DA041480. The funders had no role in study design, data collection and analysis, decision to publish or preparation of the manuscript. The authors thank Dr. James Waltz for providing an earlier version of the reversal-learning e-prime code. The authors acknowledge the help, support, and advice of Dr. Sarah Fineberg, Dr. Albert Powers III, and Dr. Pantelis Leptourgos.

## Additional information

### Funding

| Funder | Grant reference number | Author |
|---|---|---|
| NIMH | R01MH12887 | Philip R Corlett |
| NIMH | R21MH120799-01 | Stephanie Mary Groman Philip R Corlett |
| International Mental Health Research Organization | Janssen Rising Star Translational Research Award | Philip R Corlett |
| Interacting Minds Centre | Pilot Project Award | Philip R Corlett |
| NIH | Medical Scientist Training Program Training Grant | Erin J Reed |
| NIH | GM007205 | Erin J Reed |
| NINDS | Neurobiology of Cortical Systems Grant | Erin J Reed |
| NINDS | T32 NS007224 | Erin J Reed |
| Gustavus and Louise Pfeiffer Research Foundation | Fellowship | Erin J Reed |
| NSF | DGE1122492 | Stefan Uddenberg |
| NSF | DGE1752134 | Stefan Uddenberg |
| NIDA | DA DA041480 | Stephanie Mary Groman |

The funders had no role in study design, data collection and interpretation, or the decision to submit the work for publication.

### Author contributions

Erin J Reed, Conceptualization, Data curation, Formal analysis, Investigation, Visualization, Methodology, Writing - original draft, Writing - review and editing; Stefan Uddenberg, Software, Writing - review and editing; Praveen Suthaharan, Formal analysis, Visualization, Writing - review and editing; Christoph D Mathys, Software, Formal analysis, Writing - original draft, Writing - review and editing; Jane R Taylor, Resources, Supervision, Writing - review and editing; Stephanie Mary Groman, Conceptualization, Resources, Software, Formal analysis, Supervision, Methodology, Writing - review and editing; Philip R Corlett, Conceptualization, Resources, Supervision, Funding acquisition, Validation, Investigation, Visualization, Methodology, Writing - original draft, Project administration, Writing - review and editing

## Author ORCIDs

Erin J Reed (iD) https://orcid.org/0000-0003-1669-1929
Stephanie Mary Groman (iD) http://orcid.org/0000-0002-5231-0612
Philip R Corlett (iD) https://orcid.org/0000-0002-5368-1992

## Ethics

Human subjects: Experiments were conducted at Yale University and the Connecticut Mental Health Center (New Haven, CT) in strict accordance with Yale University's Human Investigation Committee and Institutional Animal Care and Use Committee. Informed consent was provided by all research participants (Yale HIC# 2000022111: Beliefs and Personality Traits).

Animal experimentation: This study was performed in strict accordance with the recommendations in the Guide for the Care and Use of Laboratory Animals of the National Institutes of Health. All of the animals were handled according to approved institutional animal care and use committee (IACUC) at Yale University.

## Decision letter and Author response

Decision letter https://doi.org/10.7554/eLife.56345.sa1
Author response https://doi.org/10.7554/eLife.56345.sa2

# Additional files

## Supplementary files

• Transparent reporting form

## Data availability

Data are available on ModelDB83 (http://modeldb.yale.edu/258631) with accession code p2c8q74m. Figures 2-10 have associated raw data available. Code for the HGF toolbox v5.3.1 is freely available at https://translationalneuromodeling.github.io/tapas/.

The following dataset was generated:

| Author(s) | Year | Dataset title | Dataset URL | Database and Identifier |
|---|---|---|---|---|
| Reed EJ, Uddenberg S, Suttaharan P, Mathys CD, Taylor JR, Groman SM, Corlett PR | 2020 | Paranoia as a deficit in non-social belief updating | http://modeldb.yale.edu/258631 | ModelDB, p2c8q74m |

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
