## [Decision Letter]

**Acceptance summary:**

In this study, the authors tested the ability of humans and rats to track probabilities of reward in a 3-option discrimination task. Paranoia/meth use was associated with worse performance on the task, reflected in fewer reversals in humans and increases in suboptimal win-switch, lose-stay responding, and these tendencies were associated with an increase in the model parameter reflecting phasic volatility and a reduction in the model parameter reflecting contextual volatility. The authors conclude that alterations in perceptions of environmental volatility – uncertainty – may play a significant causal role in paranoia.

**Decision letter after peer review:**

Thank you for submitting your article "A paranoid style of belief updating across species" for consideration by *eLife*. Your article has been reviewed by three peer reviewers, including Geoffrey Schoenbaum as the Reviewing Editor and Reviewer #1, and the evaluation has been overseen by Floris de Lange as the Senior Editor.

The reviewers have discussed the reviews with one another and the Reviewing Editor has drafted this decision to help you prepare a revised submission.

We would like to draw your attention to changes in our policy on revisions we have made in response to COVID-19 (https://elifesciences.org/articles/57162). Specifically, when editors judge that a submitted work as a whole belongs in *eLife* but that some conclusions require a modest amount of additional new data, as they do with your paper, we are asking that the manuscript be revised to either limit claims to those supported by data in hand, or to explicitly state that the relevant conclusions require additional supporting data.

Summary:

In this study, the authors tested the ability of humans and rats to track probabilities of reward in a 3-option discrimination task. Performance was challenged outright reversal of reward probabilities of the different options (phasic volatility) as well as a shift in the spread of probabilities across blocks (contextual volatility). The effect of different types of volatility on performance were modeled and correlated with paranoia in the humans and with effects of methamphetamine in the rats, the use of which has been associated with paranoia in humans. Paranoia/meth use was associated with worse performance on the task, reflected in fewer reversals in humans and increases in suboptimal win-switch, lose-stay responding, and these tendencies were associated with an increase in the model parameter reflecting phasic volatility and a reduction in the model parameter reflecting contextual volatility. The authors conclude that alterations in perceptions of environmental volatility – uncertainty – may play a significant causal role in paranoia. Overall the reviewers agreed that the paper addressed an important question using an exciting combination of behavior and computational models, and that the results were compelling and potentially important. The main concerns revolved around a desire for more clarity in the presentation and some effort to contrast the current results with other possible models.

Essential revisions:

Key areas of revision were threefold. Together these encompass most of the individual reviewer remarks, which are left below to be addressed rather than reproduced here.

1) Two of the reviewers found it difficult to follow some of the logic and explanations. So the most important revision is to make the specific points raised in the reviews more clear while at the same time simplifying the results to be more digestible for readers who are not computational modelers. This might include removing some experiments, showing more data initially, etc.

2) Remove the rat experiment – it does not really match the others and the paper will be much simpler without it. Of course if the authors disagree, this is their prerogative but in this case it needs to be better explained why the differences are not important. For instance, it is somewhat unclear how the rat behavior can effectively model context volatility as it does not include this in the design. It could also be included as supplemental perhaps.

3) Two reviewers questioned whether the model used is superior to simpler models in interpreting the behavior. Some comparison would be useful to show that the three level model applied is superior.

Reviewer #1:

In this study, the authors tested the ability of humans and rats to track probabilities of reward in a 3-option discrimination task. Performance was challenged outright reversal of reward probabilities of the different options (phasic volatility) as well as a shift in the spread of probabilities across blocks (contextual volatility). The effect of different types of volatility on performance were modeled and correlated with paranoia in the humans and with effects of methamphetamine in the rats, the use of which has been associated with paranoia in humans. Paranoia/meth use was associated with worse performance on the task, reflected in fewer reversals in humans and increases in suboptimal win-switch, lose-stay responding, and these tendencies were associated with an increase in the model parameter reflecting phasic volatility and a reduction in the model parameter reflecting contextual volatility. The authors conclude that alterations in perceptions of environmental volatility – uncertainty – may play a significant causal role in paranoia.

Overall I really like the use of the task variants and modeling to identify links between paranoia and simple learning parameters. I did find it hard to decipher some of the Results sections and the modeling. I think the paper would benefit greatly from being written with more up front handholding for readers who are not well-versed in these concepts. This might be accomplished by laying out more clearly how the different parameters can be understood both intuitively to impact learning/paranoia as well as how they are directly related to behavior in the tasks. This might include presenting more of the behavioral data. Currently all that is presented are the model parameters. It would be more convincing I think if the actual performance was shown from the subjects and then from the model, along with the parameters.

As part of this, I also am not sure the rodent data really fits. I like its inclusion in principle, but the task does not correspond directly to the variants used in humans. Specifically it lacks the shift in context volatility. This seems crucial to me. I think perhaps it might be removed to simplify the presentation. Likewise the task variants that do not include this could be removed.

On an interpretive level, I had two further questions. The first is whether it is possible to reproduced the performance with simpler models? Or how much of an improvement is gained with the use of the more complex model? Beyond this I also wonder if the authors believe that some of the effects might be compensatory – that is if I undertand correctly, they are arguing that there is less of an impact of context volatility on behavior in the experimental subjects. If this is true, it seems to me it might lead to more surprise when there are sudden changes in reward probability when a reversal occurs….?

Reviewer #2:

The authors ran 2 experiments in human subjects (one in the lab, the other online) and re-analysed behavioural data in rats and found that: 1) in humans, paranoid scores are correlated with an impairment in volatility monitoring according to a Bayesian meta-learning framework: 2) in rats meta-amphetamine administration (a pharmacological manipulation that induces paranoia in humans) impairs uncertainty monitoring in a similar way. Overall, I liked this paper; I think it represents an important contribution.

My main questions / suggestions are about the choice model-free metrics and statistical analyses and computational modeling inferences.

1) In Experiment 1, the difference between high/low paranoia is on the “number of reversals” variable. In Experiment 2 (and in the rats) the difference between high/low paranoia (placebo/ meta-amphetamine) is captured by the “win / switch” rate. However, I could not find the “win / switch” rate measure for Experiment 1 and the “number of reversal” metric for Experiment 2. The authors should report the same behavioural metrics for all experiments.

2) Even if expected, the correlation between depression, anxiety and paranoia is a bit annoying. I am convinced that paranoia is the main determinant of the computational effects, but I think the authors could provide some additional evidence that this is the case. A possible solution could be to use a structural equation modeling. Another (possibly better) solution would be to run a PCA on the three scales (the average scores, not necessarily the individual items): my prediction is that the first component will have positive loadings on the three scales and the second will be specific to the paranoia scale. They could then correlate the PCA values instead of the scores of the scales.

3) I think that, in addition to the current model, the authors could also test a simple RL model with different learning rates for positive and negative prediction errors (see Lefebvre et al., NHB, 2017). I think the readers would be interested in knowing these results as the learning rate asymmetry has been shown to correlate with the striatum, as meta-amphetamine affects dopamine and also because in paranoia there seems to be an affective component to paranoia. This analysis could be done in parallel (not in antagonism) of the main model and reported in the SI.

Reviewer #3:

This study takes on the hypothesis that paranoia is actually due to dysfunction in recognizing volatility. They address this through two human experiments (one in-person comparing individuals with and without psychiatric diagnoses; one online using Amazon Mechanical Turk) and a rat experiment in which rats are exposed to methamphetamine or saline. They justify their claims by fitting a model using a Hierarchical Gaussian Filter (HGF) and identifying changes in the underlying parameters, particularly identifying larger priors for higher volatility in the high paranoia group and in the methamphetamine-exposed rats.

The strength of this paper is that it uses a simple task to explore important topics, particularly a transdiagnostic perspective. However, we have several serious problems with the manuscript, including both the communication and the experiments and analysis itself. While we laud the authors for attempting to compare experiments across species, we do not find the rat experiment a good parallel for the human.

Major concerns:

– Overall, it was very difficult to read this paper. Even with multiple read-throughs each and multiple discussions between the reviewers (senior π and graduate student), we are not sure that we understand the manuscript, its goals, or its conclusions. Many of the figures are not referenced in the text (Tables 5 and 9 are never referenced in the paper at all), many of the figures are unclear as to their purpose (what is being plotted in Figure 4?), and many of the figures are very poorly explained (we finally concluded that we are supposed to track colors not position in Figure 3). A careful use of supplemental figures, a better track to the storyline, and better communication overall would improve this paper dramatically.

– The rat experiment is interesting, but it is not a good comparison for the human data. The rat experiment is within-subject, comparing pre and post-manipulation, while the human data is between-subject, comparing high and low paranoia scores. Furthermore, the experiment itself is completely different. While the human experiments had three decks that changed throughout, the rats had two changing and one unchanging deck. We recommend removing the rat experiment.

Unclear concerns

– These Bayesian models (such as the HGF shown in Figure 2) are notoriously unstable. Very small changes can produce dramatically different results. How independent are these variables? Are there other models that can be compared?

– The authors seem to be trying to make the argument that the real issue with paranoia is not the social decision-making process, but rather an underlying issue with measuring volatility (and particularly meta-volatility). As such, the title of the paper should be changed. The important part of this paper (as we understand it) is not the cross-species translation (which is problematic at best), but rather the new model of paranoia as a dysfunction elsewhere than the social sphere.

– The authors need to add citations for using rats exposed to methamphetamine as a model for paranoia. While there appears to be research supporting this method, the authors do not actually cite it. To our knowledge, It is not appropriate to describe methamphetamine as a locus coeruleus disruption or as a change in the noradrenergic gain. Yet, the discussion about the rat experiment seems to be based on noradrenergic gain manipulations.

Specific concerns

– Figure 1: what is the difference between performance independent and performance dependent changes? Explain in figure caption.

– Figure 2B: Once we finally realized that the key to this figure were the colors, we liked that the authors kept the colors consistent across the rats and humans, since the rat comparisons were pre-Rx instead of having two blocks, and therefore likely indistinguishable from the low-paranoia group pre-Rx. However, it makes comparison of the figures confusing, because we expect the comparisons across the figures to mean the same thing to compare the outcomes. This figure requires much better and clearer explanations.

– Figure 3: Is there a reason that the authors expected version 3 to be significant over version 4? Why might the order of context change matter (or not matter)?

– Figure 4: This figure is confusing and at the moment does not provide additional understanding of the results. Consider relabeling and adjusting figure caption to explain what is in the figure and move the results to the Results section or display in a table (or both), or otherwise remove it in total.

– Figure 5 seems important, but shouldn't we see this for all of the important variables? We thought the argument was primarily about metavolatility rather than phasic volatility coupling.

– Figure 7 is important, but was poorly labeled and mostly impenetrable. In particular, panel a is completely unclear. There are no labels, no explanation for colors or any other components. What is the difference between simulated and "recovered" parameters?

– Figure 8 claims a replication between in-person and online, but the online appear significant, while the in-person do not.

– Figures 9 and 10 are impenetrable. What is this cluster analysis and how is it done?

– Figure 11 seems more appropriate to a supplemental figure.

[Editors' note: further revisions were suggested prior to acceptance, as described below.]

Thank you for resubmitting your work entitled "A paranoid style of belief updating across species" for further consideration by *eLife*. Your revised article has been evaluated by Floris de Lange (Senior Editor) and a Reviewing Editor.

The manuscript has been improved but there are some remaining issues that need to be addressed before acceptance. In particular, while the concerns of reviewer 1 and 2 were addressed, and the manuscript is markedly improved in terms of clarity, reviewer 3 still has some remaining requests. They are described below.

Reviewer #1:

The authors have addressed my concerns.

Reviewer #2:

The authors successfully addressed my concerns.

Reviewer #3:

Overall, the paper is much clearer in its explanation of the design, analyses, and simulations. The authors have also made a clearer argument for including the rat data. However, we believe they still need to explicitly state the limitations of the human and rat experiments. Additionally, the graphs still need to be brought up to the level of clarity of the writing (particularly Figure 7). In summary, the authors have successfully clarified many questions about the analyses and conclusions of the paper, yet additional work is needed surrounding the rat vs. human experimental comparisons.

Major concerns:

– The title needs to be changed, as it is misleading regarding the findings. What seems to be the main argument is that paranoia-like behaviors are evident in belief-updating outside of a social lens, so perhaps something clearer could be something about how paranoia may arise from belief-updating, rather than social cognition. For example, we recommend a title such as "Paranoia may arise from general problems in belief-updating rather than specific social cognition" or something like that.

– Add a paragraph outlining the limitations of cross-species comparison, particularly the fact that the rats are compared within subject, while the humans are compared between subjects.

– The social nature of rats is heavily debated, and while we know they are not as social as humans, there may be some sociality for the rats. Nevertheless, the task is still asocial, and therefore assists the argument of the paper. However, if the authors are going to discuss that rats are asocial animals, we think they should include a paragraph discussing the support for and against this statement, and relate it to the asocial nature of the task. Along with this argument, please mention how rats were housed, which speaks to their sociality.

– Figure 7 has not been adequately addressed from the first round of revisions. It is still largely impenetrable. For instance, what is the left side of 7A? What are the lines? Can you describe what "choice trajectory" is? Phrases like "the purple shaded errobars indicate…" would be very helpful to the reader.

– In general, the figures need more work. Fonts are too small (particularly Figure 4), which makes it difficult to really interpret the graphs. Along with that, many of the graphs have a lot of panels and not a lot of text to describe what each of the panels means. Thorough explication of the figures would improve the paper tremendously.

---

## [Author Response]

Reviewer #1:In this study, the authors tested the ability of humans and rats to track probabilities of reward in a 3-option discrimination task. Performance was challenged outright reversal of reward probabilities of the different options (phasic volatility) as well as a shift in the spread of probabilities across blocks (contextual volatility). The effect of different types of volatility on performance were modeled and correlated with paranoia in the humans and with effects of methamphetamine in the rats, the use of which has been associated with paranoia in humans. Paranoia/meth use was associated with worse performance on the task, reflected in fewer reversals in humans and increases in suboptimal win-switch, lose-stay responding, and these tendencies were associated with an increase in the model parameter reflecting phasic volatility and a reduction in the model parameter reflecting contextual volatility. The authors conclude that alterations in perceptions of environmental volatility – uncertainty – may play a significant causal role in paranoia.Overall I really like the use of the task variants and modeling to identify links between paranoia and simple learning parameters. I did find it hard to decipher some of the Results sections and the modeling. I think the paper would benefit greatly from being written with more up front handholding for readers who are not well-versed in these concepts. This might be accomplished by laying out more clearly how the different parameters can be understood both intuitively to impact learning/paranoia as well as how they are directly related to behavior in the tasks. This might include presenting more of the behavioral data. Currently all that is presented are the model parameters. It would be more convincing I think if the actual performance was shown from the subjects and then from the model, along with the parameters.

We are glad that the reviewer liked our work. We agree that some of our presentation could be made more accessible to readers with different backgrounds in modeling experience. In the much-revised version of the paper, we signpost the modeling results much more clearly. As requested, we show model simulations next to behavioral data.

As part of this, I also am not sure the rodent data really fits. I like its inclusion in principle, but the task does not correspond directly to the variants used in humans. Specifically it lacks the shift in context volatility. This seems crucial to me. I think perhaps it might be removed to simplify the presentation. Likewise the task variants that do not include this could be removed.

We take the point. The paper was unwieldy. As we argue above, we would prefer not to remove either these rat data or these data from the task variants.

The different task variants serve as important controls for our volatility manipulation in version 3 – the task version that increased uncertainty about the task in a manner that distinguished the high from the low paranoia participants.

With regards to these rat data – we now state more explicitly what the differences are between the tasks – and what the similarities are. The reviewer is correct, we do not manipulate the volatility context in the rodent task like we did in version 3 of the human task. However, we believe these rat data are still key and informative. When confronted with increased task volatility, even low-paranoia participants began to behave more stochastically. The high paranoia participants evinced this stochasticity, even before the contextual shift towards higher volatility, during the easy task blocks. The easy task blocks are more similar to the contingencies that the rats experienced. Chronic exposure to methamphetamine made rats behave similarly to high-paranoia humans on this comparable contingency. We believe that is worth reporting. It supports further exploration of this task in a rodent setting, with all of the tools available to computational behavioral neuroscience, in order to better understand and ultimately treat paranoia.

On an interpretive level, I had two further questions. The first is whether it is possible to reproduced the performance with simpler models? Or how much of an improvement is gained with the use of the more complex model?

In response to this reviewer and the others, we fit simpler models. Those simpler models did not capture the behavioral effects or group differences in our data and as such, we conclude that our three-layer model is the most appropriate.

Beyond this I also wonder if the authors believe that some of the effects might be compensatory – that is if I undertand correctly, they are arguing that there is less of an impact of context volatility on behavior in the experimental subjects. If this is true, it seems to me it might lead to more surprise when there are sudden changes in reward probability when a reversal occurs….?

This is an interesting thought. To phrase it differently, might the increased expectation of volatility in paranoid participants be adaptive somehow? In the unprecedented uncertainty that we are currently experiencing, people who had prepared all along (and been ridiculed for it) might feel validated. There are others – expertly reviewed Raihani and Bell^2^ – who have evoked “the smoke detector principle”^3,4^ to explain paranoia – that is, a series of false alarms (even if costly) is preferable to a catastrophic miss^5^. However, our data advise against the conclusion that paranoia is an adaptive solution to high volatility. This is because in addition to high expected volatility, paranoid participants also appear impaired at learning from volatility (captured in our κ parameter), they expect volatility (captured by μ_3_^0^), but cannot use it adaptively to update their beliefs appropriately (more negative ω_3_). This would seem an extremely deleterious combination, and one which captures the broad reach of paranoia and the fact that it fails to satisfy its adherents – there is always something new to be concerned about, some new dimension that ones’ persecutors can reach into.

Reviewer #2:The authors ran 2 experiments in human subjects (one in the lab, the other online) and re-analysed behavioural data in rats and found that: 1) in humans, paranoid scores are correlated with an impairment in volatility monitoring according to a Bayesian meta-learning framework: 2) in rats meta-amphetamine administration (a pharmacological manipulation that induces paranoia in humans) impairs uncertainty monitoring in a similar way. Overall, I liked this paper; I think it represents an important contribution.My main questions / suggestions are about the choice model-free metrics and statistical analyses and computational modeling inferences.1) In Experiment 1, the difference between high/low paranoia is on the “number of reversals” variable. In Experiment 2 (and in the rats) the difference between high/low paranoia (placebo/ meta-amphetamine) is captured by the “win / switch” rate. However, I could not find the “win / switch” rate measure for Experiment 1 and the “number of reversal” metric for Experiment 2. The authors should report the same behavioural metrics for all experiments.

We are grateful for this opportunity to clarify. Average win-switch rates and numbers of reversals are reported in Tables 1 and 2. We recognize that the tables were perhaps too densely populated with information. The reporting of behavioral data is consistent between the Experiments 1 and 2, with the exception of study-specific metrics such as number of null trials, which only occurred in Experiment 1. Behavioral metrics have been previously published for Experiment 3 (see Groman et al., 2018^6^).

2) Even if expected, the correlation between depression, anxiety and paranoia is a bit annoying. I am convinced that paranoia is the main determinant of the computational effects, but I think the authors could provide some additional evidence that this is the case. A possible solution could be to use a structural equation modeling. Another (possibly better) solution would be to run a PCA on the three scales (the average scores, not necessarily the individual items): my prediction is that the first component will have positive loadings on the three scales and the second will be specific to the paranoia scale. They could then correlate the PCA values instead of the scores of the scales.

This is a great suggestion. We performed the PCA as suggested, combining these data from the SCID Paranoia questions, the Beck Depression and Beck Anxiety Inventories. The scree plot depicts the three-principle component solution. We regressed each on the kappa parameter, and only principle component 1 correlated with kappa. Unpacking the contribution of each scale to PC1, it is clear that depression, anxiety and paranoia all contribute to PC1. We suggest that this finding is consistent with the idea that depression and anxiety represent contexts in which paranoia can flourish and likewise, harboring a paranoid stance toward the world can induce depression and anxiety. We report this analysis in the revised version if the manuscript. The multiple regression that we included in the manuscript does however suggest that the relationship between paranoia and kappa is paramount, since, in that model, kappa was not related to depression or anxiety, but remained significantly related to paranoia.

3) I think that, in addition to the current model, the authors could also test a simple RL model with different learning rates for positive and negative prediction errors (see Lefebvre et al., NHB, 2017). I think the readers would be interested in knowing these results as the learning rate asymmetry has been shown to correlate with the striatum, as meta-amphetamine affects dopamine and also because in paranoia there seems to be an affective component to paranoia. This analysis could be done in parallel (not in antagonism) of the main model and reported in the SI.

Thank you. We fit this model. We find no difference in prediction error weightings between our high and low paranoia participants. This simpler model does not capture the patterns in our data. We now report this analysis in the revised paper.

Reviewer #3:This study takes on the hypothesis that paranoia is actually due to dysfunction in recognizing volatility. They address this through two human experiments (one in-person comparing individuals with and without psychiatric diagnoses; one online using Amazon Mechanical Turk) and a rat experiment in which rats are exposed to methamphetamine or saline. They justify their claims by fitting a model using a Hierarchical Gaussian Filter (HGF) and identifying changes in the underlying parameters, particularly identifying larger priors for higher volatility in the high paranoia group and in the methamphetamine-exposed rats.The strength of this paper is that it uses a simple task to explore important topics, particularly a transdiagnostic perspective. However, we have several serious problems with the manuscript, including both the communication and the experiments and analysis itself. While we laud the authors for attempting to compare experiments across species, we do not find the rat experiment a good parallel for the human.Major concerns:– Overall, it was very difficult to read this paper. Even with multiple read-throughs each and multiple discussions between the reviewers (senior π and graduate student), we are not sure that we understand the manuscript, its goals, or its conclusions. Many of the figures are not referenced in the text (Tables 5 and 9 are never referenced in the paper at all), many of the figures are unclear as to their purpose (what is being plotted in Figure 4?), and many of the figures are very poorly explained (we finally concluded that we are supposed to track colors not position in Figure 3). A careful use of supplemental figures, a better track to the storyline, and better communication overall would improve this paper dramatically.

This is a fair criticism. We had prepared our work as a paper with supplementary materials. Unfortunately, *eLife* does not permit supplementary materials and so we had to integrate them into our manuscript. This made the piece unwieldy. We have thoroughly revised the manuscript for clarity of presentation. We feel it is much improved. We hope that π and graduate student agree.

– The rat experiment is interesting, but it is not a good comparison for the human data. The rat experiment is within-subject, comparing pre and post-manipulation, while the human data is between-subject, comparing high and low paranoia scores. Furthermore, the experiment itself is completely different. While the human experiments had three decks that changed throughout, the rats had two changing and one unchanging deck. We recommend removing the rat experiment.

We respectfully disagree. There are key differences between the human and rat tasks of course, however, the rat methamphetamine manipulation captures the apparent stochasticity of the high paranoia participants even in response to the simple or easy contingency. Our data show that this stochasticity arises in high paranoid humans and methamphetamine exposed rats for exactly the same computational reasons. As such, we prefer to retain the rat experiment. In the much-revised manuscript, which we hope is much clearer, we now emphasize the task differences so that readers are aware of them.

Unclear concerns– These Bayesian models (such as the HGF shown in Figure 2) are notoriously unstable. Very small changes can produce dramatically different results. How independent are these variables? Are there other models that can be compared?

In response to these reviewers and all other reviewers, we computed simpler models (one reinforcement learning model and one simpler HGF model). Those models failed to capture task induced and group differences that we sought to explain. Taken together with the fact that our chosen model yields parameters which, when used to simulate data, recapitulate the win-switching and stochastic behavior we observed in high paranoia, we believe that the model is the most appropriate. We hope that our more careful and clear unpacking of our modeling approach and our data is more interpretable and understandable.

The only choices the HGF modelling results are sensitive to are those of the priors of the estimated parameters. While the choice of priors affects the model’s performance, the nature of this effect is different from that seen in chaotic systems, which the reviewer seems to be referring to. In our model, small changes to priors lead to small changes in estimated parameters and inferred belief trajectories.

– The authors seem to be trying to make the argument that the real issue with paranoia is not the social decision-making process, but rather an underlying issue with measuring volatility (and particularly meta-volatility). As such, the title of the paper should be changed. The important part of this paper (as we understand it) is not the cross-species translation (which is problematic at best), but rather the new model of paranoia as a dysfunction elsewhere than the social sphere.

We agree, this paper is about volatility processing as a mechanism for paranoia, relatively free from the social domain (which has been the focus for most paranoia research in humans). We disagree that the cross-species part is problematic, as we have outlined above. In fact, the rodent data is an important key-stone of our argument. Compared to human and non-human primates, rodents are relatively asocial. They are also free of the socioeconomic factors often associated with paranoia. The observation of similar behaviors in a similar task under the influence of manipulations relevant to human paranoia bolsters our argument that the observed “style” of learning dysfunction is not constricted to the social domain. One of the biggest take-home points and implications of non-social learning dysfunction is that future studies can explore the neural substrates of paranoia-relevant learning mechanisms in animal models without needing to emulate the complexities of paranoid social relationships. But we agree that our aim – to deliver an account of paranoia that focuses not on the social, but in basic belief updating mechanisms – could have been clearer. We now clarify in the revised manuscript.

– The authors need to add citations for using rats exposed to methamphetamine as a model for paranoia. While there appears to be research supporting this method, the authors do not actually cite it. To our knowledge, It is not appropriate to describe methamphetamine as a locus coeruleus disruption or as a change in the noradrenergic gain. Yet, the discussion about the rat experiment seems to be based on noradrenergic gain manipulations.

We now cite the extensive literature on methamphetamine’s impact on noradrenaline release and locus-coeruleus function (see for example^7-9^).

Specific concerns– Figure 1: what is the difference between performance independent and performance dependent changes? Explain in figure caption.

Thank you, we now explain in the caption that performance dependent changes elicit reversals after a certain number of correct responses, performance independent changes occur when reversals are imposed regardless of participant behavior.

– Figure 2B: Once we finally realized that the key to this figure were the colors, we liked that the authors kept the colors consistent across the rats and humans, since the rat comparisons were pre-Rx instead of having two blocks, and therefore likely indistinguishable from the low-paranoia group pre-Rx. However, it makes comparison of the figures confusing, because we expect the comparisons across the figures to mean the same thing to compare the outcomes. This figure requires much better and clearer explanations.

We have revised the legend and in-text description of this figure.

– Figure 3: Is there a reason that the authors expected version 3 to be significant over version 4? Why might the order of context change matter (or not matter)?

We thought that moving from an easy to a harder task context would be significant because the easy context would be easier to acquire than the hard. If the hard were completed first, the expectations would be weaker and so less confounded by the subsequent changes in the underlying contingencies.

– Figure 4: This figure is confusing and at the moment does not provide additional understanding of the results. Consider relabeling and adjusting figure caption to explain what is in the figure and move the results to the Results section or display in a table (or both), or otherwise remove it in total.

We have removed what was Figure 4.

– Figure 5 seems important, but shouldn't we see this for all of the important variables? We thought the argument was primarily about metavolatility rather than phasic volatility coupling.

Kappa is the parameter that replicated across all experimental contexts and survived correction for multiple statistical comparisons and correction for all the potential demographic confounders we queried. It is the parameter that we are most confident in explaining the group differences. We correlated it to paranoia as a further test of our hypothesis. We did not feel it appropriate to correlate every parameter from the model with every clinical variable. The point is that volatility learning rate and priors on volatility capture the differences between the groups and travel together in all of the three experiments we report (as evidenced by the meta-analytic p-value and cluster analyses).

– Figure 7 is important, but was poorly labeled and mostly impenetrable. In particular, panel a is completely unclear. There are no labels, no explanation for colors or any other components. What is the difference between simulated and "recovered" parameters?

We now clearly unpack the figures in their legends and in the text. Simulated parameters are those that we estimate back from simulated behavioral choices. Recovered parameters were extracted from the models fit to real behavioral data. That they correlate with one another suggests that we have an appropriate model that recapitulates behavioral choices that match those that we observed experimentally.

– Figure 8 claims a replication between in-person and online, but the online appear significant, while the in-person do not.

We replicate the broad pattern of changes in behaviors and model parameters across the three experiments – as evidenced by the meta-analytic p-value analysis and the cluster analysis. The results are not necessarily completely identical but they are highly consistent across the studies.

– Figures 9 and 10 are impenetrable. What is this cluster analysis and how is it done?

We now unpack the cluster analysis more clearly in the manuscript.

– Figure 11 seems more appropriate to a supplemental figure.

We have removed this figure.

[Editors' note: further revisions were suggested prior to acceptance, as described below.]

The manuscript has been improved but there are some remaining issues that need to be addressed before acceptance. In particular, while the concerns of reviewer 1 and 2 were addressed, and the manuscript is markedly improved in terms of clarity, reviewer 3 still has some remaining requests. They are described below.Reviewer #3:Overall, the paper is much clearer in its explanation of the design, analyses, and simulations. The authors have also made a clearer argument for including the rat data. However, we believe they still need to explicitly state the limitations of the human and rat experiments.

We now explicitly state the limitations as requested, acknowledging both the design differences and the broader debate about sociality in rats:

“There are some important limitations to our conclusions. Compared with humans, rats are relatively asocial. But they are not completely asocial. In our experiment they were housed in pairs, and, more broadly, they evince social affiliative interactions with other rats. A further limitation centers on the comparability of our experimental designs. In humans our comparisons were both within (contingency transition) and between groups (low versus high paranoia). In rats, the model was also mixed with some between (saline versus methamphetamine) and some within-subject (pre versus post chronic treatment) comparisons. We should be clear that there was no contingency context transition in the rat study. However, just as that transition made low paranoia humans behave like high paranoia humans, chronic methamphetamine exposure made rats behave on a stable contingency much like high paranoia humans – even in the absence of contingency transition.”

Additionally, the graphs still need to be brought up to the level of clarity of the writing (particularly Figure 7). In summary, the authors have successfully clarified many questions about the analyses and conclusions of the paper, yet additional work is needed surrounding the rat vs. human experimental comparisons.

We included vector (PDF) files of the figures, which should be clearer than the embedded figures. We feel the best way to improve clarity is with more detailed figure legends, which we now include, with a particular emphasis on Figure 7.

We agree that some of the figures were dense and perhaps distracting. For example, Figure 10 was intended to be a supplementary figure depicting our control analyses for the clustering. As we noted previously, *eLife* does not allow supplementary figures. On the basis of the reviewer’s comments, we opted to remove Figure 10 and describe the results of the control analyses in the text. We deemed another large multi-paneled figure to be surplus to requirements.

Major concerns:– The title needs to be changed, as it is misleading regarding the findings. What seems to be the main argument is that paranoia-like behaviors are evident in belief-updating outside of a social lens, so perhaps something clearer could be something about how paranoia may arise from belief-updating, rather than social cognition. For example, we recommend a title such as "Paranoia may arise from general problems in belief-updating rather than specific social cognition" or something like that.

While we disagree that the title is misleading, we have changed the title at the reviewer’s request. We chose:

“Paranoia as a deficit in non-social belief updating”

– Add a paragraph outlining the limitations of cross-species comparison, particularly the fact that the rats are compared within subject, while the humans are compared between subjects.

Per above, this has been noted as a limitation in the Discussion:

“There are some important limitations to our conclusions. Compared with humans, rats are relatively asocial. But they are not completely asocial. In our experiment they were housed in pairs, and, more broadly, they evince social affiliative interactions with other rats. A further limitation centers on the comparability of our experimental designs. In humans our comparisons were both within (contingency transition) and between groups (low versus high paranoia). In rats, the model was also mixed with some between (saline versus methamphetamine) and some within-subject (pre versus post chronic treatment) comparisons. We should be clear that there was no contingency context transition in the rat study. However, just as that transition made low paranoia humans behave like high paranoia, chronic methamphetamine exposure made rats behave on a stable contingency much like high paranoia humans – even in the absence of contingency transition.”

– The social nature of rats is heavily debated, and while we know they are not as social as humans, there may be some sociality for the rats. Nevertheless, the task is still asocial, and therefore assists the argument of the paper. However, if the authors are going to discuss that rats are asocial animals, we think they should include a paragraph discussing the support for and against this statement, and relate it to the asocial nature of the task.

Again, per above, this statement has been made.

Along with this argument, please mention how rats were housed, which speaks to their sociality.

We now state:

“Compared with humans, rats are relatively asocial. But they are not completely asocial. In our experiment they were housed in pairs, and, more broadly, they evince social affiliative interactions with other rats^1-3^”

Rats were housed in pairs. This has also been noted:

“In our experiment they were housed in pairs,”

– Figure 7 has not been adequately addressed from the first round of revisions. It is still largely impenetrable. For instance, what is the left side of 7A? What are the lines? Can you describe what "choice trajectory" is? Phrases like "the purple shaded errobars indicate…" would be very helpful to the reader.

We are sorry that the figure was largely impenetrable. This type of figure is typically included in supplementary files. However, as noted previously, *eLife* does not allow supplementary figures. We include the figure in order to highlight how actual participant choices and inferred beliefs (following our observing the observer approach) as well as beliefs inferred from simulated choices (themselves grounded in perceptual parameters estimated from actual behavior) are very similar. It is encouraging that the kappa parameter (which captures the impact of phasic volatility on belief updating, survives correction for multiple comparisons, replicates across studies in its association with paranoia and paranoia-relevant states and drives clustering) is well recovered.

We now breakdown the legend panel by panel, describing each feature.

We hope that it is now clearer.

Here is the new legend for Figure 7:

“Figure 7. Parameter recovery. a, Actual subject trajectory: this is an example choice trajectory from one participant (top). The layers correspond to the three layers of belief in the HGF model (depicted in Figure 2A). Focusing on the low-level beliefs (yellow box): The purple line represents the subject’s estimated first-level belief about the value of choosing deck 1; blue, their belief about the value of choosing deck 2; and red, their belief about the value of choosing deck 3. Simulated subject trajectory represents the estimated beliefs from choices simulated from estimated perceptual parameters from that participant (middle), and Recovered subject trajectory represents what happens when we re-estimate beliefs from the simulated choices (bottom). Crucially, Simulated trajectories closely align with real trajectories (the increases and decreased in estimated beliefs about the values of each deck [purple, blue, red lines] align with each other across actual, simulated and recovered trajectories), although trial-by-trial choices (colored dots and arrow) occasionally differ. Outcomes (1 or 0; black dots and arrows) remain the same. b, Actual versus Recovered: these data represent the belief parameters estimated from the participant’s responses (Actual) compared to those estimated from the choices simulated with the participant’s perceptual parameters (Recovered). Actual and Recovered values significantly correlate for 𝛚_2_ (r=0.702, p=2.52E-11) and 𝛋 (r=0.305, p=0.011) but not 𝛚_3_ (r=0.172, p=0.16) or 𝛍_3_^0^ (r=0.186, p=0.13). Box plots: gray indicates low paranoia, orange designates high paranoia; center lines depict medians; box limits indicate the 25th and 75th percentiles; whiskers extend 1.5”

– In general, the figures need more work. Fonts are too small (particularly Figure 4), which makes it difficult to really interpret the graphs. Along with that, many of the graphs have a lot of panels and not a lot of text to describe what each of the panels means. Thorough explication of the figures would improve the paper tremendously.

Per our response above, we have included PDF vector files that can be readily enlarged. We feel the panels are all necessary. The best way to improve the figures, we believe, is to increase the detail included in the legends, which we now do throughout. We also removed Figure 10. Its many panels and complexity ultimately distracted from the simple message that the cluster analysis is robust to removal of various halves of the data.

References

1 Palminteri, S., Wyart, V. & Koechlin, E. The Importance of Falsification in Computational Cognitive Modeling. *Trends Cogn Sci* 21, 425-433, doi:10.1016/j.tics.2017.03.011 (2017).

2 Raihani, N. J. & Bell, V. An evolutionary perspective on paranoia. *Nat Hum Behav* 3, 114-121, doi:10.1038/s41562-018-0495-0 (2019).

3 Nesse, R. M. The smoke detector principle: Signal detection and optimal defense regulation. *Evol Med Public Health* 2019, 1, doi:10.1093/emph/eoy034 (2019).

4 Nesse, R. M. The smoke detector principle. Natural selection and the regulation of defensive responses. *Ann N Y Acad Sci* 935, 75-85 (2001).

5 Green, M. J. & Phillips, M. L. Social threat perception and the evolution of paranoia. *Neurosci Biobehav Rev* 28, 333-342, doi:10.1016/j.neubiorev.2004.03.006 (2004).

6 Groman, S. M., Rich, K. M., Smith, N. J., Lee, D. & Taylor, J. R. Chronic Exposure to Methamphetamine Disrupts Reinforcement-Based Decision Making in Rats. *Neuropsychopharmacology* 43, 770-780, doi:10.1038/npp.2017.159 (2018).

7 Ferrucci, M. et al. The Effects of Amphetamine and Methamphetamine on the Release of Norepinephrine, Dopamine and Acetylcholine From the Brainstem Reticular Formation. *Front Neuroanat* 13, 48, doi:10.3389/fnana.2019.00048 (2019).

8 Ferrucci, M., Giorgi, F. S., Bartalucci, A., Busceti, C. L. & Fornai, F. The effects of locus coeruleus and norepinephrine in methamphetamine toxicity. *Curr Neuropharmacol* 11, 80-94, doi:10.2174/157015913804999522 (2013).

9 Ferrucci, M., Pasquali, L., Paparelli, A., Ruggieri, S. & Fornai, F. Pathways of methamphetamine toxicity. *Ann N Y Acad Sci* 1139, 177-185, doi:10.1196/annals.1432.013 (2008).